# Nicotine engages a VTA-NAc feedback loop to inhibit amygdala-projecting dopamine neurons and induce anxiety-like behaviors

Tinaïg Le Borgne [1,2], Claire Nguyen [2], Eléonore Vicq[1,2], Joachim Jehl [1,2], Clément Solié [1,2], Nicolas Guyon [1], Louison Daussy[1], Aylin Gulmez[1], Lauren M. Reynolds [1,2], Sarah Mondoloni [2], Stéfania Tolu [2,3], Stéphanie Pons[4], Uwe Maskos [4], Emmanuel Valjent [5], Alexandre Mourot [1,2], Philippe Faure [1,2,6] ✉ & Fabio Marti [1,2,6] ✉

Nicotine activates ventral tegmental area (VTA) dopaminergic (DA) neurons projecting to the nucleus accumbens (NAc) to drive its reinforcing effects. Simultaneously, nicotine inhibits those projecting to the amygdala (Amg) to mediate anxiety-like behavior through a process that remains unknown. Here, we show that in male mice, NAc- and Amg-projecting DA neurons respond with similar polarities to ethanol and nicotine, suggesting a shared network-based mechanism underlying the inhibitory effect of these otherwise pharmacologically-distinct drugs. Selective activation of NAc-projecting DA neurons, using genetic or optogenetic strategies, produced inhibition of Amg-projecting DA neurons, through a GABAergic feedback loop. Furthermore, optogenetically silencing this feedback loop prevented nicotine from inducing both inhibition of DA neurons and anxiety-like behavior. Therefore, nicotine-induced inhibition of the VTA-Amg DA pathway results from a VTA-NAc inhibitory feedback loop, mediating anxiety-like behavior.

Dopamine (DA) neurons in the ventral tegmental area (VTA) of the reward system play a crucial role in addiction mechanisms. Drugs of abuse, including nicotine and alcohol, act on this reward system through a variety of cellular mechanisms, ultimately leading to DA release in the nucleus accumbens (NAc) and behavioral reinforcement[1–3]. Accordingly, global optogenetic activation of VTA DA neurons can induce conditioned place preference[4] or reinforcement in a self-administration paradigm[5], whereas inhibition of the same neurons induces avoidance behavior[6]. However, VTA DA neurons exhibit remarkable heterogeneity in their molecular properties and projection areas[7–9]. These neurons not only encode appetitive stimuli and reward predictions, but also salient signals such as aversive or

alarming events[10–12]. Understanding the heterogeneity of drug effects on these neurons is therefore important to grasp the multifaceted nature of drug abuse.

Nicotine acts on nicotinic acetylcholine receptors (nAChRs), a family of ligand-gated ion channels[13], and increases the activity of both DA and GABA neurons in the VTA[3,14]. Our previous work has shown that nicotine not only activates VTA DA neurons as traditionally described, but also induces inhibition of a subpopulation of these neurons[15,16]. VTA DA neurons activated by nicotine project to the NAc and their optogenetic activation induces reinforcement, while those inhibited by nicotine project to the amygdala (Amg) and their optogenetic inhibition mediates anxiety-like behavior[16]. VTA DA neurons can thus mediate both the

[1]Plasticité du Cerveau CNRS UMR8249, École supérieure de physique et de chimie industrielles de la Ville de Paris (ESPCI Paris), Paris, France. [2]Neuroscience Paris Seine CNRS UMR 8246 INSERM U1130, Institut de Biologie Paris Seine, Sorbonne Université, Paris, France. [3]Unité de Biologie Fonctionnelle et Adaptative, CNRS, F-75013, Université Paris Cité, Paris, France. [4]Unité Neurobiologie intégrative des systèmes cholinergiques, Département de neuroscience, Institut Pasteur, Paris, France. [5]Institut des Neurosciences de Montpellier, Université de Montpellier, INSERM U1298, Montpellier, France. [6]These authors jointly supervised this work: Philippe Faure, Fabio Marti. ✉e-mail: phfaure@gmail.com; fabio.marti@sorbonne-universite.fr

rewarding and anxiogenic properties of nicotine. Understanding how neurons projecting to Amg are inhibited by nicotine is, however, a major challenge in studying the dual effects of nicotine on reward and emotional circuits. We have previously shown a complete abolition of nicotine responses - both activation and inhibition−in mice deleted for the β2 nAChR subunit (β2[-/-] mice), and restoration of both types of response following constitutive re-expression of the β2 subunit in the VTA (i.e., in all neuronal populations)[16]. These results indicate that nicotine activation and inhibition both stem from nicotine's local action in the VTA. However, since nAChRs are cation-conducting ion channels, nicotine exerts a depolarizing action on neurons expressing these receptors. It is therefore unlikely that nicotine directly inhibits any neuronal population. The observed nicotine-induced inhibition most likely involves a network effect arising from the activation of VTA cells expressing β2-containing nAChRs (β2*nAChRs).

Interestingly, alcohol, like nicotine, induces both activation and inhibition of distinct neuronal populations in the VTA[17], suggesting that this pattern of opposite responses is not restricted to nicotine and may result from a circuit-based mechanism. Despite employing different molecular mechanisms[18], these two drugs are known to increase DA release in the NAc by enhancing the activity of VTA DA neurons[16,19]. However, it remains unknown whether alcohol, like nicotine, inhibits a specific DA pathway, and whether these two drug-induced inhibitions share common features. In the present study, we investigated the

mechanism by which nicotine and alcohol inhibit specific VTA DA neurons and induce anxiety-like behavior.

## Results

### Both nicotine and alcohol induce excitation and inhibition of distinct VTA DA neuron subpopulations in vivo

We first investigated whether nicotine and alcohol, two drugs with different molecular modes of action, could elicit similar responses in VTA DA neurons. To this end, in vivo single-cell juxtacellular recordings and labeling were performed in anesthetized mice to record the activity of VTA DA neurons during consecutive intravenous (i.v.) injection of nicotine (Nic; 30 μg/kg) and ethanol (EtOH; 250 mg/kg). All recorded neurons were confirmed as DA neurons by post hoc immunofluorescence with co-labeling for tyrosine hydroxylase and neurobiotin (TH+, NB+; Fig. 1a). Acute i.v. paired injections of nicotine and ethanol induced a significant increase or decrease in the firing rate (Fig. 1b, c), illustrated by bimodal distribution of firing frequency variations across neurons that was absent in vehicle injections (Fig. S1A, B). Among the 72 neurons recorded, 69 exhibited a significant response to nicotine while 67 neurons showed a significant response to ethanol compared to baseline variation (see "Methods"). Consistent with our previous work[16], nicotine induced activation (Nic+; n = 41) or inhibition (Nic−; n = 28) of different populations of DA neurons (Fig. 1d). Similarly, neurons were activated (EtOH+, n = 52) or

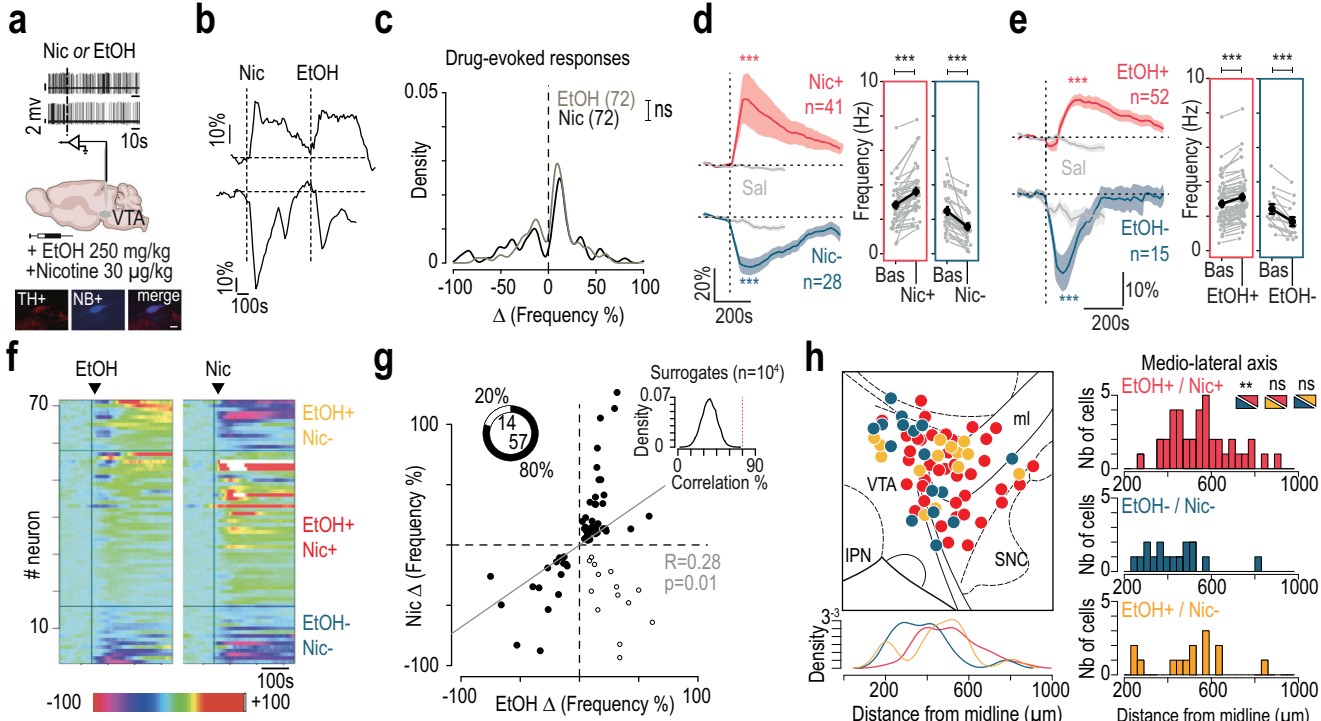

**Fig. 1 | Nicotine and ethanol induce activation and inhibition of VTA DA subpopulations. a** In vivo juxtacellular recordings of VTA dopamine (DA) neurons following intravenous (i.v.) injection of ethanol (EtOH; 250 mg/kg) or nicotine (Nic; 30 μg/kg). Recorded neurons were labeled with neurobiotin (NB) and identified via tyrosine hydroxylase (TH) immunofluorescence. Scale bar: 20 μm. **b** Examples of normalized firing frequency change after paired Nic and EtOH injections for activated (top) and inhibited (bottom) DA neuron. **c** Distribution of firing rate responses (% change) to nicotine (black, n = 72) or ethanol (gray, n = 72); Kolmogorov-Smirnov test, D = 0.14, p = 0.49. **d** Left: Mean time course of normalized firing frequency after nicotine or saline injection in activated (Nic+, red, n = 41) and inhibited (Nic-, blue, n = 28) neurons (Maximum % variation, paired Wilcoxon: Nic+ vs Sal, V = 854, ***p = 1.7e[-11]; Nic- vs Sal, V = 3, ***p = 3.7e[-08]). Right: individual firing rate variation from baseline (Nic+ vs Baseline, V = 861, ***p = 2.5e[-08]; Nic- vs Baseline, V = 0, ***p = 7.4e[-09]). **e** Same as (**d**) for ethanol. Ethanol-induced activation

(EtOH+, red, n = 52) and inhibition (EtOH-, blue, n = 15) differed significantly from saline (paired Wilcoxon: EtOH+ vs Sal, V = 1290, ***p = 4.5e[-08;] EtOH- vs Sal, V = 3, ***p = 0.0003) and from baseline (EtOH+ vs Bas, V = 1378, ***p = 3.6e[-10]; EtOH- vs Bas, V = 0, ***p = 6.1e[-05]). **f** Individual neuron responses to ethanol (left) and nicotine (right), ranked from most inhibited (blue) to most activated (white/red). Neurons grouped into EtOH-/Nic-, EtOH+/Nic+, and EtOH+/Nic-. **g** Correlation between EtOH- and Nic-induced responses (Pearson's r = 0.28, t_69 = 2.45, *p = 0.02). Correlated responses (black dots, n = 57); uncorrelated (white dots, n = 14). Inset: experimental (red line) vs surrogate correlation distributions. **h** Left: Anatomical distribution of EtOH+/Nic+ (red, n = 39), EtOH-/Nic- (blue, n = 16), and EtOH+/Nic- (orange, n = 14) neurons. Right: EtOH-/Nic- neurons were more medially located than EtOH+/Nic+ neurons (Wilcoxon, V = 472, **p = 0.0028); EtOH+/Nic- locations were not significantly different (V = 147, p = 0.14; V = 312, p = 0.44). Data are presented as mean ± SEM. All statistical tests are two-sided.

inhibited (EtOH−, $n = 15$) by ethanol (Fig. 1e). Furthermore, ethanol-activated and ethanol-inhibited neurons were anatomically segregated along the medio-lateral axis of the VTA (Fig. S1C, D), mirroring the spatial organization observed with nicotine[16].

When comparing the individual responses of each neuron to nicotine and ethanol injections (Fig. 1f), we found that 80% of the neurons responded with similar polarity to both substances (i.e., activated or inhibited by both drugs; Fig. 1g). Strikingly, all neurons inhibited by ethanol were also inhibited by nicotine (EtOH-/Nic-, 15/15), and all neurons activated by nicotine were also activated by ethanol (EtOH+/Nic+, 42/42), suggesting a shared mechanism of action. However, nearly half of the neurons inhibited by nicotine were activated by ethanol (EtOH+/Nic-, 14/29), indicating an opposing effect of the two drugs on these neurons. Anatomical mapping reveals that EtOH-/Nic- and EtOH+/Nic+ neurons were spatially segregated along the medio-lateral axis within the VTA (Fig. 1h and S1D), suggesting they constitute two distinct subpopulations. In contrast, EtOH+/Nic- neurons were not distinctly localized from the others within the VTA (Fig. 1h), implying they may constitute a less lateralized third subpopulation.

Overall, these findings indicate that the effects of nicotine and ethanol on DA neurons are highly dependent on anatomical organization. The polarity of a neuron's response−whether inhibition or activation−is an intrinsic property rather than dose-dependent, as ethanol consistently induced the same response polarity across all doses tested (Fig. S1H, I), similar to previous findings with nicotine[16]. Finally, we examined how individual and combined administration of nicotine and ethanol affected neuronal activity (Fig. S1J). Co-injection induced a greater response than either drug alone, indicating that ethanol and nicotine act additively rather than competitively and do not saturate each other, at least at the tested doses.

### Nicotine and alcohol activate VTA DA neurons projecting to the NAc, while they inhibit those projecting to the Amg

To confirm that inhibited and activated neurons belong to distinct DA pathways, we injected retrobeads (RB) into either the Amg (BLA, CeA; Fig. S2A, B) or the NAc (NAcLSh, NAcMSh, NAcCore; Fig. S2C, D), followed by in vivo single-cell juxtacellular recordings during paired injection of the two drugs. Triple immunofluorescence labeling enabled post hoc confirmation of the DA nature (TH+) and the projection site (RB+; Fig. 2a). As expected, anatomical segregation along the medio-lateral axis was observed between NAc- and Amg-projecting DA neurons (Fig. S2E).

Consistent with previous findings following i.v. nicotine injection[16], NAc-projecting DA neurons were mostly activated (12/13) whereas Amg-projecting DA neurons were predominantly inhibited (15/16), as indicated by the distribution of maximum frequency variation from baseline (Fig. 2b, c) and the mean response relative to i.v. saline injection (Fig. 2d). Ethanol i.v. injection produced a similar dichotomy in responses between NAc-projecting and Amg-projecting VTA DA neurons (Fig. 2e). Most of the NAc-projecting DA neurons were activated (11/13), while most of the Amg-projecting DA neurons were inhibited (13/16) by i.v. ethanol injection (Fig. 2f, g). Finally, the polarity of frequency variations induced by the two drugs was correlated in 92.3% of NAc-projecting and 87.5% of Amg-projecting VTA DA neurons (Fig. 2h). Notably, the proportion of EtOH-/Nic+ neurons previously observed (Fig. 1g) dropped from 20% to 7%, suggesting that most of these neurons project to targets other than NAc or Amg.

These findings indicate that nicotine and ethanol produce similar patterns of activation and inhibition in these distinct dopaminergic pathways. Specifically, it suggests a common network mechanism by which both drugs inhibit Amg-projecting DA neurons.

### NAc-projecting DA neurons are more sensitive to nicotine ex vivo

Nicotine binds to nAChRs expressed by DA and GABA neurons in the VTA[20], directly exciting DA neurons and indirectly inhibiting them through activation of GABA neurons[14]. To understand the mechanism underlying the inhibition, we investigated the balance between direct excitatory and indirect inhibitory effects of nicotine on DA neurons projecting to the NAc (LSh) or the Amg (BLA) using ex vivo patch-clamp recordings in VTA slices. We first examined nicotine-induced excitatory currents by applying nicotine puffs to neurons identified by co-labeling of TH, RB and biocytin (Fig. 3a). Nicotine induced smaller currents in Amg-projecting DA neurons compared to NAc-projecting DA neurons at all doses tested (10, 30 and 100 μM; Fig. 3b). This indicates that NAc-projecting DA neurons, which are known to be activated by nicotine in vivo, are more responsive to nicotine than Amg-projecting DA neurons, and suggests a differential expression of nAChRs in these two neuronal subpopulations.

We then investigated whether nicotine-induced inhibition could be mediated by a difference in local VTA GABAergic inter-neurons. Nicotine (30 μM) was perfused locally on brain slices and spontaneous or nicotine-evoked inhibitory postsynaptic currents (IPSCs) were recorded within the two subpopulations of DA neurons. In Amg-projecting VTA DA neurons, which are known to be inhibited in vivo, nicotine had no detectable effect on either the amplitude or frequency of IPSCs (Fig. 3c, *top*; Fig. S3A). Conversely, NAc-projecting VTA DA neurons showed a significant increase in IPSC amplitude following nicotine application (Fig. 3c, *bottom*), without a corresponding change in frequency (Fig. S3A). In addition, bath application of nicotine to the slice evoked a long-lasting nicotinic inward current of larger amplitude in NAc-projecting DA neurons compared to Amg-projecting ones (Fig. S3B), in line with our observations using nicotine puffs (Fig. 3b). These results indicate that NAc-projecting DA neurons exhibit greater responsiveness to nicotine, both through nAChR-mediated excitatory effect and inhibitory effect triggered by the activation of VTA GABAergic interneurons[14]. These observations are consistent with previous results demonstrating the necessity of co-activating DA neurons and GABAergic interneurons by nicotine for DA release in the NAc[14]. It also suggests that, at the level of the VTA, nicotine triggers distinct GABAergic signaling for DA neurons projecting to the NAc and Amg. Importantly, without nicotine and under TTX (recordings of miniature IPSCs), both the mean amplitude and frequency of events were comparable between the two populations (Fig. S3C, D), indicating similar overall GABAergic innervation.

Thus, the distinct nicotine-induced GABAergic signaling observed in NAc- versus Amg-projecting DA neurons likely results from active modulation of local interneuron dynamics rather than from structural differences in GABAergic connectivity. Taken together, these results also suggest that nicotine-induced inhibition of Amg-projecting DA neurons is unlikely to be mediated primarily by an increased GABAergic signaling from local VTA interneurons.

### Nicotine-induced activation of NAc-projecting DA neurons is sufficient to trigger inhibition of VTA DA neurons

Our ex vivo investigation revealed that NAc-projecting DA neurons show a stronger activation in response to directly-applied nicotine than Amg-projecting DA neurons. Therefore, we explored whether the inhibition of Amg-projecting DA neurons could result from the activation of NAc-projecting DA neurons. Initially, we tested whether the selective activation of NAc-projecting DA neurons by nicotine could lead to the inhibition of VTA DA neurons in vivo. In $\beta2^{-/-}$ mice, a retrograde virus expressing a Cre recombinase was injected into the three subareas of the NAc (NAcLSh, NAcMSh, NAcCore) and a Cre-dependent lentivirus expressing the β2 subunit was injected into the VTA. This approach restricted the expression of the β2 subunit in NAc-projecting neurons. We then performed in vivo juxtacellular recordings of VTA DA neurons in vectorized mice expressing either GFP alone, as control (GFP$^{NAcVec}$), or β2-GFP (β2$^{NAcVec}$; Fig. 3d). In β2$^{NAcVec}$ mice, i.v. nicotine induced firing variations characterized by a multimodal distribution different from

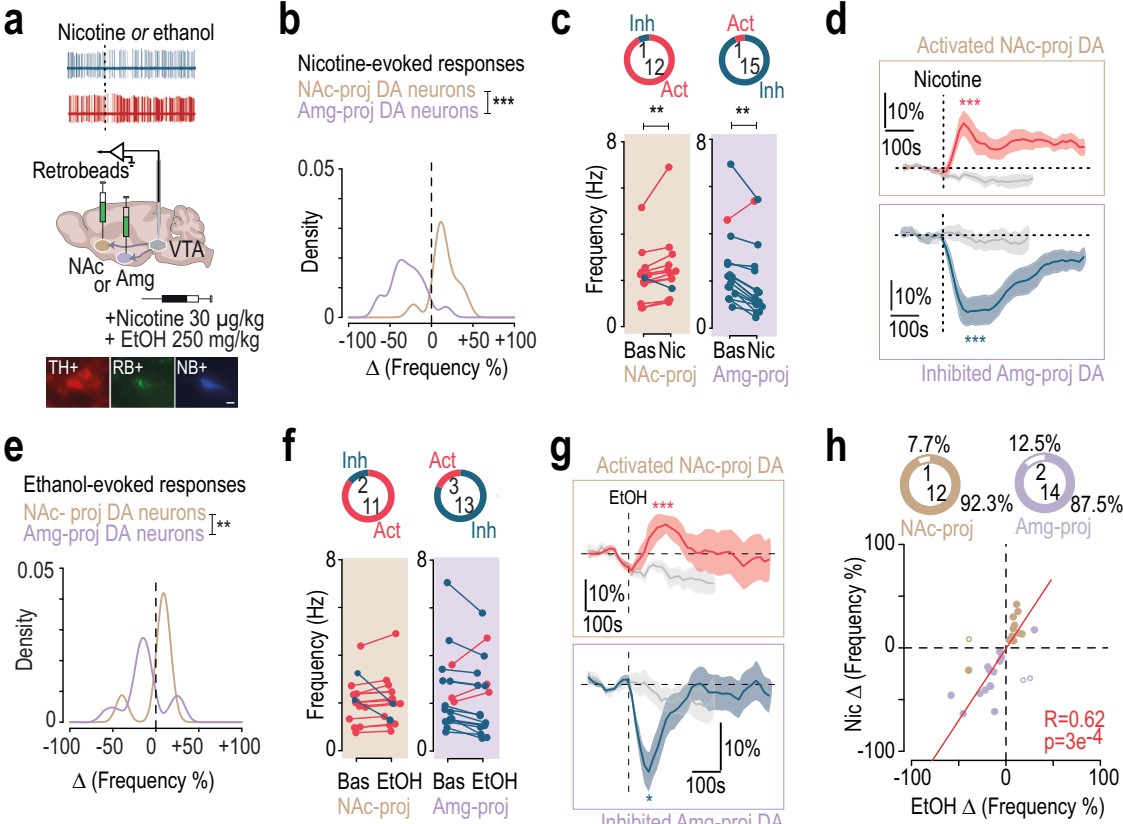

**Fig. 2 | Nicotine and ethanol induce similar response profiles across two VTA DA pathways. a** In vivo juxtacellular recordings were performed following retrobead injections into either the nucleus accumbens (NAc: lateral shell, medial shell, core) or the amygdala (Amg: basolateral and central nuclei). VTA dopamine (DA) neurons responding to i.v. ethanol (EtOH; 250 mg/kg) or nicotine (Nic; 30 μg/kg) were identified post hoc via immunofluorescence for tyrosine hydroxylase (TH), neurobiotin (NB), and retrobeads (RB). Scale bar: 20 μm. **b** Distribution of nicotine-evoked firing rate changes (% of baseline) in NAc- (brown, $n = 13$) and Amg-projecting (purple, $n = 16$) DA neurons (Kolmogorov-Smirnov test: D = 0.86, ***$p = 8.4e^{-06}$). **c** Left: proportion and individual firing rate variation from baseline of activated/inhibited NAc-projecting DA neurons after nicotine (Wilcoxon test: V = 80, **$p = 0.01$). Right: same for Amg-projecting neurons (V = 11, **$p = 0.002$). **d** Mean time course of normalized firing frequency after nicotine in activated NAc-

projecting (top, $n = 12$) and inhibited Amg-projecting (bottom, $n = 15$) neurons, compared to saline (Maximum of % variation, t-test: NAc-proj, $t_{11} = 4.58$, ***$p = 0.0007$; Amg-proj, $t_{13} = -6.26$, ***$p = 2.9e^{-05}$). **e, f** Same analyses as in (**b, c**) after ethanol. Ethanol-induced firing rate changes were significantly different between NAc- and Amg-projecting DA neurons (Kolmogorov-Smirnov test: D = 0.66, **$p = 0.002$). **g** same as (**d**) for ethanol in activated NAc-projecting (top, $n = 11$) and inhibited Amg-projecting (bottom, $n = 13$) DA neurons (Maximum of % variation, t-test: NAc-proj, $t_{11} = 4.83$, ***$p = 0.0005$; Amg-proj, $t_{12} = -2.34$, *$p = 0.04$). **h** Ethanol- and nicotine-induced responses were significantly correlated across all DA neurons (Pearson's correlation: $t_{27} = 4.15$, R = 0.62, ***$p = 0.0003$). Correlated (filled dots) and uncorrelated (empty dots) responses are shown for NAc- ($n = 12/1$) and Amg-projecting ($n = 14/2$) neurons. Data are presented as mean$n ± $ SEM. All statistical tests are two-sided.

saline, which was not observed in GFP^NAcVec control mice (Fig. 3e). Peaks in the distribution of responses observed in β2^NAcVec mice indicate varying degrees of inhibition and activation, likely reflecting differential β2 subunit re-expression within the VTA DA neuron population. While VTA DA neurons of GFP^NAcVec mice did not respond to nicotine, both nicotine-induced activation and inhibition were restored in β2^NAcVec mice (Fig. 3f). Following the same strategy, we also selectively re-expressed the β2 subunit in the VTA-Amg pathway of β2^-/- mice (β2^AmgVec; Fig. 3g). In these mice, nicotine-induced responses exhibited a distribution characterized by a peak centered at zero and a smaller peak at ~30% increase, which was not observed with saline, but was not significantly different from that observed in GFP^AmgVec mice (Fig. 3h). Nicotine-activated neurons showed a firing rate variation statistically different from saline injection, but no significant inhibition of DA neurons in response to nicotine was observed in β2^AmgVec mice (Fig. 3i). Thus, specific nicotine-induced activation of NAc-projecting, but not Amg-projecting DA neurons, was sufficient to restore nicotine-induced inhibition in a subset of VTA DA neurons.

Together, these results reveal that nicotine-induced inhibition is a consequence of the activation of VTA-NAc projections. Such a

mechanism of inhibition could arise from the activation of D2 auto-receptors following somatodendritic DA release from VTA DA neurons that have been activated by nicotine. To investigate this possibility, we examined in vivo responses to nicotine in VTA DA neurons from DAT-D2R^-/- mice, which specifically lack D2 receptors in DA neurons. Both excitatory and inhibitory responses were unaffected in these mice, indicating that D2 autoreceptors do not underlie nicotine-induced inhibition (Fig. S4).

Overall, these results indicate that nicotine-induced inhibition does not rely on local GABAergic circuitry or DA release. Consequently, we proceeded to test whether it may instead be triggered by the activation of VTA-NAc projections, which in turn engage long-range GABAergic neurons targeting the VTA.

**Amg-projecting DA neurons are targeted by inhibitory inputs from the NAc**

NAc medium spiny neurons expressing dopaminergic D1 receptors (D1-MSNs) are known to send direct inhibitory projections to VTA neurons[21–23]. First, we examined the connectivity between NAc GABAergic terminals in the VTA and BLA-projecting DA

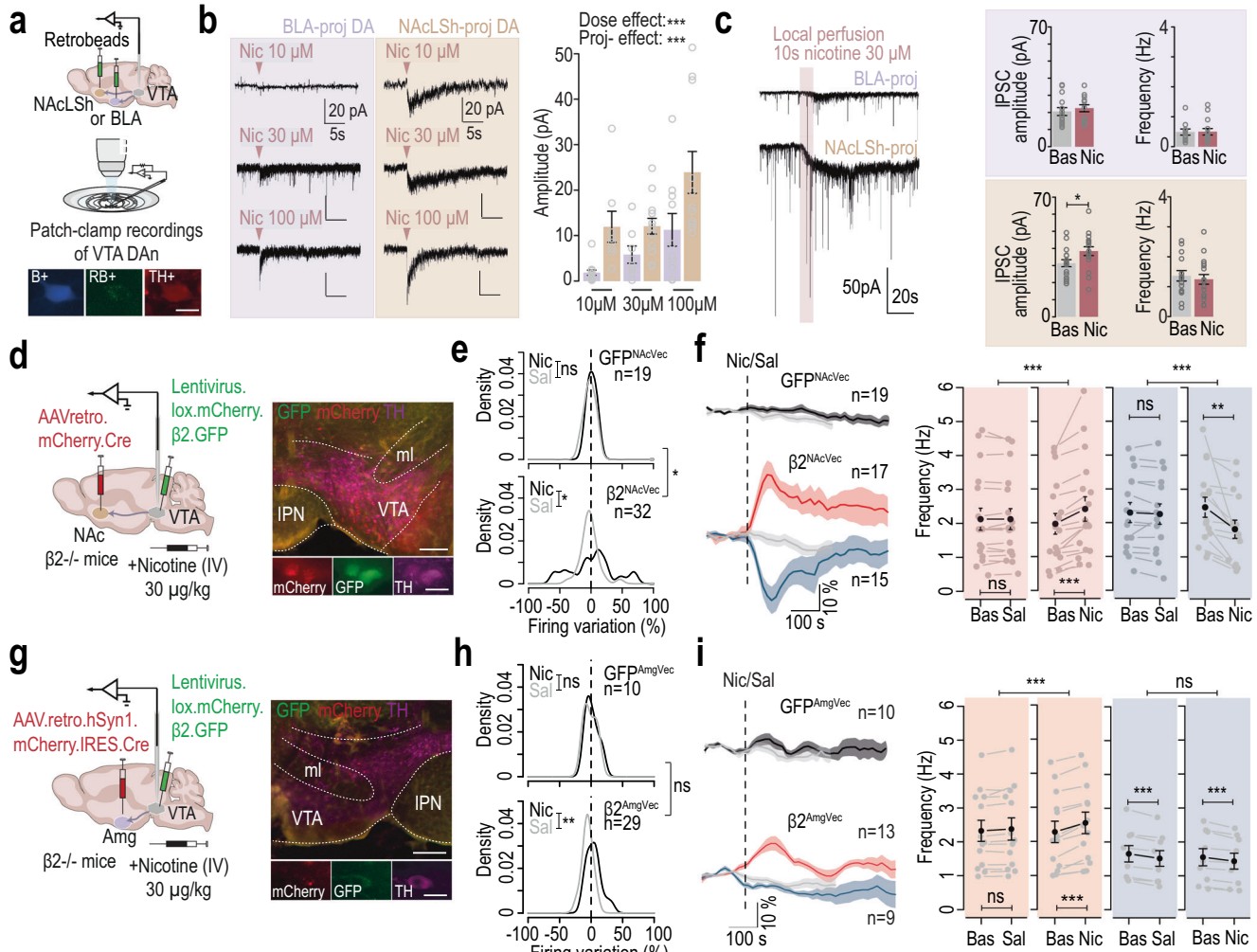

**Fig. 3 | Nicotine-induced activation of NAc-projecting neurons triggers inhibition in the VTA. a** Retrobeads injection into the nucleus accumbens lateral shell (NAcLSh) or the basolateral amygdala (BLA). Patch-clamp recordings of DA neurons identified by tyrosine hydroxylase (TH, red), neurobiotin (NB, blue), and retrobeads (RB). Scale bar: 20 μm. **b** Left: Nicotine-evoked currents in NAcLSh- and BLA-projecting DA neurons by puffs at 10, 30, and 100 μM. Right: Mean current amplitudes in NAcLSh-projecting (brown: $n = 8/13/11$) and BLA-projecting (purple: $n = 13/8/10$) neurons (RM ANOVA: dose effect, $F_{2,57} = 9.54$, ***$p = 0.0002$; projection effect, $F_{1,57} = 16.77$, ***$p = 0.0001$). **c** Left: IPSCs and nicotinic currents before/after nicotine perfusion. Right: Mean IPSC amplitude and frequency pre- and post-nicotine in BLA-projecting ($n = 14$) and NAcLSh-projecting ($n = 17$) neurons (t-test: $t_{30} = -2.14$, *$p = 0.04$). **d** AAVretro.mCherry.Cre injection in NAc (lateral/medial shell and core) for Cre-dependent β2-nAChR re-expression in VTA. Left: TH, mCherry and GFP expression in VTA. Scale bars: 200 and 20 μm. **e** Firing rate responses (% change) to nicotine (30 μg/kg, black) or saline (gray) in DA neurons from GFP$^{NAcVec}$ ($n = 19$) or β2$^{NAcVec}$ ($n = 32$) mice in vivo (Kolmogorov-Smirnov: β2$^{NAcVec}$, D = 0.38, *$p = 0.02$; nicotine responses in β2$^{NAcVec}$ vs GFP$^{NAcVec}$, D = 0.38, *$p = 0.05$). **f** Left: Mean time course of normalized firing frequency following nicotine or saline in GFP$^{NAcVec}$ and β2$^{NAcVec}$. Right: individual firing rate variation for nicotine-activated (Nic+, $n = 17$, red) and inhibited (Nic-, $n = 15$, blue) neurons compare to baseline (paired t-test, Nic +, $t_{16} = -4.46$, ***$p = 0.0004$; Nic-, $t_{14} = 3.46$, **$p = 0.004$) and saline (paired Wilcoxon, Nic+, V = 0, ***$p < 0.001$; Nic-, V = 1, ***$p = 0.0001$). **g–i** Same as (**d–f**) with AAVretro injection into the amygdala (BLA/CeA). **h** GFP$^{AmgVec}$ ($n = 10$), β2$^{AmgVec}$ ($n = 29$; D = 0.45, **$p = 0.005$). Comparison with baseline (paired t-test, Nic+, $n = 13$, $t_{12} = -6.99$, ***$p = 1.5e^{-05}$; Sal-, $t_8 = 7.52$, ***$p = 6.8e^{-05}$; Nic-, $n = 9$, $t_8 = 5.14$, ***$p = 0.0008$) and saline (paired Wilcoxon, Nic+, V = 91, ***$p = 0.0002$; Nic-, $t_8 = 0.87$, $p = 0.4$). Data are presented as mean ± SEM. All statistical tests are two-sided.

neurons, by combining optogenetics with patch-clamp recordings (Fig. 4a). ChR2 was expressed in the three subareas of the NAc (in different batches of mice), red retrobeads were injected into the BLA, and light-evoked IPSCs were recorded in BLA-projecting DA neurons. Neurons were then filled with biocytin to verify their DA phenotype (Fig. 4b). We found that a significant percentage of BLA-projecting DA neurons responded to light stimulation from the NAcMSh (41.2%, $n = 7/17$ neurons), while a higher proportion responded to light stimulation from the NAcLSh (61.6%, $n = 8/13$ neurons) and almost all responded to light stimulation from the NAcCore (90.9%, $n = 11/12$ neurons; Fig. 4c). These data show that VTA DA neurons projecting to the BLA receive strong GABAergic inhibition from the NAc Shell and Core.

## NAc-projecting DA neurons trigger inhibitory feedback on VTA DA neurons

In light of these results, we sought to test whether activation of the VTA-NAc dopaminergic pathway (VTA$_{DA}$-NAc) leads to inhibition of VTA neurons by recruiting descending GABAergic inputs from the NAc. To investigate the functional impact of this pathway, we injected a retrograde ChR2-containing virus (retroChR2-YFP) in the three subareas of the NAc of DATcre mice, for restricted expression of ChR2 in VTA DA neurons projecting to the NAc (Fig. 4d, *left*). All YFP-positive cells recorded in patch-clamp experiments followed 10 and 20 Hz stimulation patterns (Fig. 4d, *middle*). Recorded neurons were confirmed as DA by post-hoc immunofluorescence verification (TH+, YFP+, B+). Neurons expressing the YFP signal were localized in the medial and

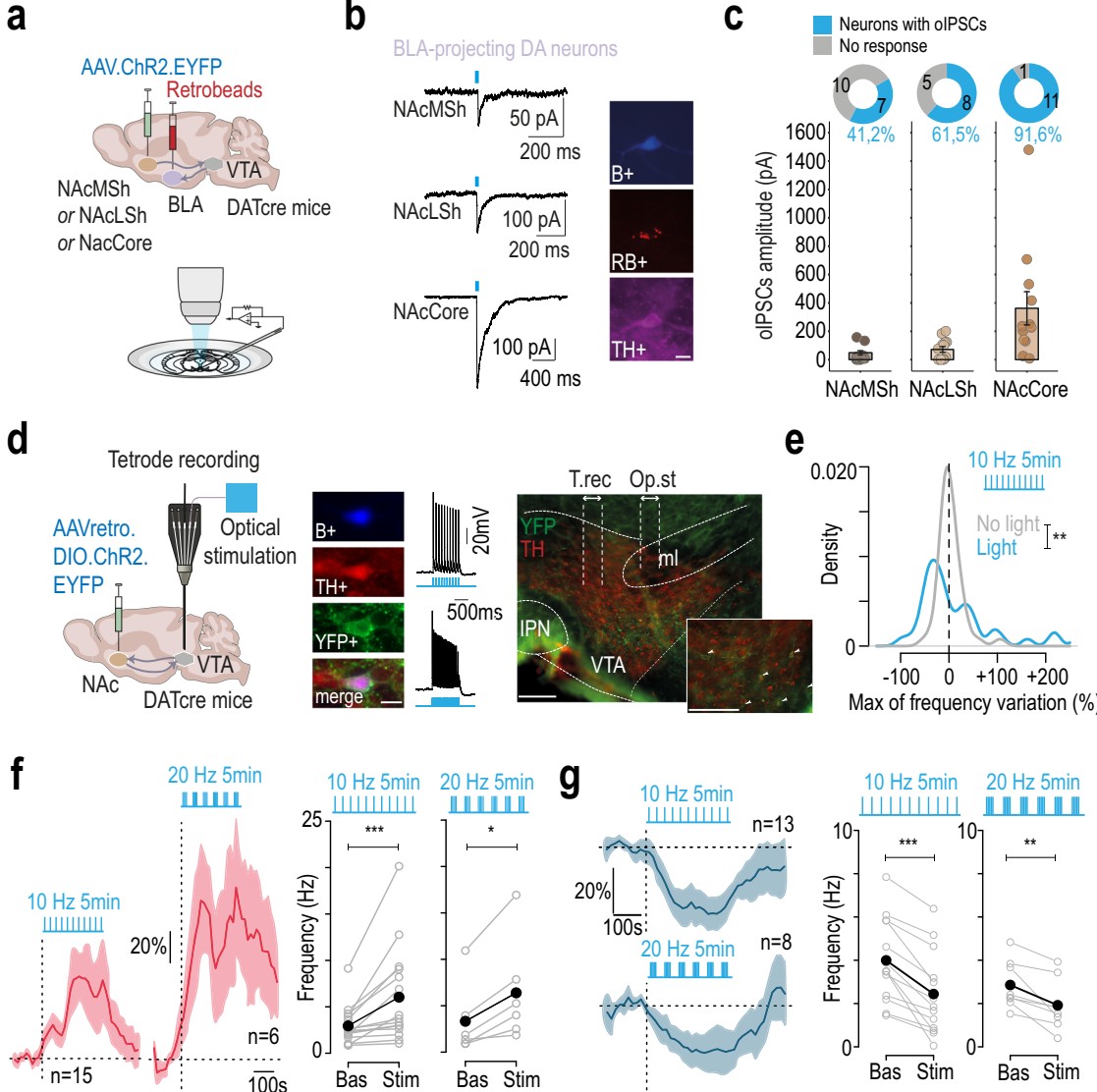

**Fig. 4 | VTA$_{DA}$-NAc pathway activates inhibitory feedback targeting VTA-BLA neurons and inducing inhibition in the VTA. a** Retrobeads were injected into the basolateral amygdala (BLA), and ChR2 was expressed in the nucleus accumbens (NAc), specifically in the lateral shell (NAcLSh), medial shell (NAcMSh), or core (NAcCore). **b** Representative light-evoked IPSCs recorded from BLA-projecting VTA dopamine (DA) neurons. Neurons were identified post hoc via co-labeling of tyrosine hydroxylase (TH), neurobiotin (NB), and retrobeads (RB). Scale bar: 20 μm. **c** Proportion of BLA-projecting VTA DA neurons exhibiting light-evoked IPSCs (oIPSCs) following optogenetic stimulation from NAcMSh, NAcLSh, or NAcCore inputs. Mean oIPSC amplitudes are shown for responsive neurons ($n = 7/17$, 8/13, and 11/12, respectively). **d** Left: A Cre-dependent, retrogradely transported AAV encoding ChR2 was injected into the NAc (NAcLSh + NAcMSh + NAcCore) of DAT-Cre mice. Patch-clamp or multi-unit extracellular recordings were then performed.

Middle: YFP-positive cells recorded ex vivo responded to 10 Hz and 20 Hz photo-stimulation. Scale bar: 20 μm. Right: Representative viral expression in the VTA and schematic of fiber (Op.st) and tetrode (T.rec) placement during in vivo recordings. Scale bars: 200 μm. **e** Distribution of changes in firing frequency (%) evoked by 10 Hz light stimulation (blue) compared to baseline spontaneous change of activity (gray) in VTA pDA neurons ($n = 43$; Kolmogorov-Smirnov test: D = 0.4, **$p = 0.002$). **f** Left: Mean time course of normalized firing frequency in light-activated pDA neurons during 10 Hz regular and 20 Hz burst photostimulation. Right: Firing rate change from baseline for regular and burst patterns ($n = 15$ and 6; paired Wilcoxon test: 10 Hz, V = 0, ***$p = 6.1e^{-05}$; 20 Hz, V = 0, *$p = 0.03$). **g** same as (**f**) for light-inhibited pDA neurons ($n = 13$ and 8, respectively; paired t-test: 10 Hz, $t_{12} = 6.1$, ***$p = 5.3e^{-05}$; 20 Hz, $t_7 = 3.87$, **$p = 0.006$). Data are presented as mean ± SEM. All statistical tests are two-sided.

lateral parts of the VTA (Fig. 4d, *right*), consistent with previous findings[8,23].

We then performed multi-unit extracellular recordings combined with optogenetic stimulation in the VTA to assess the network-level effects of selectively activating NAc-projecting DA neurons. To this end, the optical fiber was positioned laterally in the VTA (0.6–0.7 mm lateral to midline), to preferentially photoactivate NAc-projecting DA neurons, which were found to be predominantly lateralized (Fig. S2,[16]), while the tetrode was positioned medially (0.3–0.4 mm lateral to midline) to preferentially record Amg-projecting DA neurons (Fig. 4d, *right*). To mimic nicotine-induced

excitation, we applied 5 min of optogenetic stimulation using either a regular (10 Hz) or burst-like (20 Hz) pattern. These manipulations produced a multimodal distribution of firing frequency changes in putative DA (pDA) neurons, distinct from spontaneous baseline variability ($n = 43$, 10 Hz regular photostimulation, Fig. 4e). A subset of neurons showed robust excitation during stimulation (Fig. 4f), while another subset was significantly inhibited (Fig. 4g), consistent with a combination of direct activation and indirect inhibition. Although we did not observe reliable opto-tagging in the excited neurons (i.e., no significant increase in firing probability within 10 ms of light onset; Fig. S5B), this population likely corresponds to

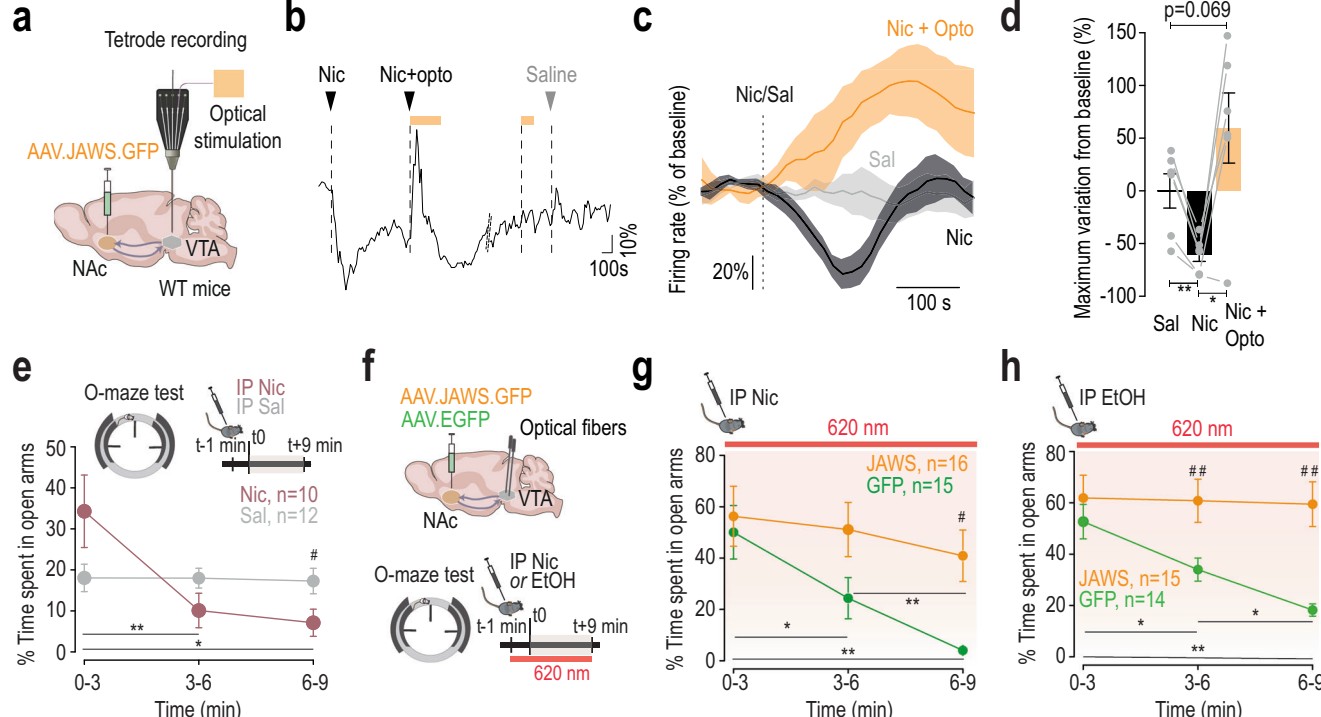

**Fig. 5 | Inhibitory feedback loop from the NAc mediates nicotine-induced inhibition of VTA pDA neurons and nicotine-induced anxiety-like behavior.**
**a** JAWS-expressing AAV injection in the NAc (NAcLShell + NAcMShell + NAcCore). Multi-unit extracellular recordings and optical stimulation in the VTA. **b** Example of firing rate change (%) after nicotine alone (Nic, i.v 30 μg/kg) or with optogenetic inhibition of NAc terminals in the VTA (Nic+Opto). **c** Mean time course of normalized firing frequency in pDA-inhibited neurons after nicotine with (orange) or without (black) NAc terminal inhibition (n = 6). **d** Firing change after saline, nicotine, or nicotine + optogenetic inhibition (n = 6). Paired t-tests: Sal vs Nic, $t_5$ = 5.76, **p = 0.007; Nic vs Nic+Opto, $t_5$ = −3.98, *p = 0.02; Sal vs Nic+Opto, $t_5$ = −2.31, p = 0.07. **e** Time spent in open arms in O-maze after nicotine (0.5 mg/kg i.p., n = 10) or saline (n = 12). RM ANOVA: interaction, $F_{2,40}$ = 8.66, p = 0.0007; time, $F_{2,40}$ = 7.81, p = 0.001. Post-hoc Wilcoxon (Holm-corrected): Nic group: 3 vs 6', V = 55,

**p = 0.006; 3 vs 9', V = 51, *p = 0.03; 6 vs 9', V = 22.5, p = 0.57; Nic vs Sal at 9', W = 94, #p = 0.03. **f** JAWS- or GFP-expressing AAV was injected in the NAc with VTA fiber implantation. O-maze test after nicotine (0.5 mg/kg) or ethanol (1 g/kg) with continuous light stimulation (260 nm). **g** Time in open arms after nicotine with NAc terminals inhibition (n = 16, 15). RM ANOVA: interaction, $F_{2,58}$ = 3.27, p = 0.04; group, $F_{1,29}$ = 4.18, p = 0.05; time, $F_{2,58}$ = 12.26, p = 3.62e$^{-05}$. Post-hoc: GFP: 3 vs 6', V = 85, *p = 0.04; 3 vs 9', V = 89, **p = 0.006; 6 vs 9', V = 90, **p = 0.006; JAWS vs GFP at 9': W = 58, #p = 0.01. **h** Same as (**g**) for ethanol (n = 15,14). interaction, $F_{2,54}$ = 4.75, p = 0.01; group, $F_{1,27}$ = 10.02, p = 0.004; time, $F_{2,54}$ = 5.94, p = 0.005. Post-hoc: GFP: 3 vs 6', V = 96, *p = 0.04; 3 vs 9', V = 102,**p = 0.007; 6 vs 9', V = 95, *p = 0.04; JAWS vs GFP: at 6', W = 44, ##p = 0.008; at 9', W = 35, ##p = 0.002. Data are presented as mean ± SEM. All statistical tests are two-sided.

ChR2-expressing, NAc-projecting DA neurons. The lack of opto-tagging may be due to the spatial lateral offset (~300 μm) between the optic fiber and the recording tetrode, which reduces the efficiency of direct photostimulation[24,25]. Consequently, we could not definitively confirm that photoactivated neurons were ChR2-positive and NAc-projecting. In parallel, the inhibited neurons were likely modulated indirectly via network effects, consistent with the inhibitory feedback from NAc to Amg-projecting DA neurons. To explore whether light-inhibited neurons correspond to Amg-projecting DA neurons, we tested their responses to i.v. nicotine injection. Of the six light-responsive neurons, five were inhibited by regular photostimulation, and four of these were also inhibited by nicotine (Fig. S5C), suggesting they may project to the Amg. These results provide compelling evidence that selective activation of NAc-projecting DA neurons, independent of nicotine, is sufficient to suppress activity in a distinct subset of VTA DA neurons.

Taken together, these results suggest that activation of NAc-projecting DA neurons leads to subsequent activation of MSNs in the NAc. Since these MSNs have established connections with BLA-projecting VTA DA neurons, they may be involved in a VTA$_{DA}$-NAc feedback inhibitory loop. This loop, triggered by nicotine or ethanol, may explain the observed inhibitory effects on a specific DA subpopulation.

## Inhibitory feedback from the NAc mediates nicotine-induced inhibition of DA neurons

To confirm that nicotine-induced inhibition involves descending fibers from the NAc to the VTA, we expressed the halorhodopsin JAWS in the three subregions of the NAc of wild-type (WT) mice, to block any NAc feedback onto the VTA (Fig. 5a). We then recorded the response of VTA pDA neurons to an i.v. injection of nicotine, focusing on neurons inhibited by the drug, while simultaneously photoinhibiting NAc terminals in the VTA. We found that in most cases, photoinhibition reversed nicotine-induced inhibition (Fig. 5b–d). Specifically, compared to saline, maximum variations in firing rate from baseline showed a significant inhibition when nicotine was injected alone, but not when combined with photoinhibition of NAc terminals, where 5 out of 6 neurons showed an increase in firing rate (Fig. 5d; Fig. S5E). Photoinhibition alone had no effect on firing frequency, suggesting that the change in nicotine's effect during light is not due to disinhibition (Fig. S5E). Similar experiments on nicotine-activated neurons reveal that photoinhibition of NAc terminals did not prevent nicotine-induced activation of DA neurons (Fig. S5F, G). However, the dose-dependent activation of DA neurons by nicotine may be modulated by NAc projections and should be further investigated. Thus, photoinhibition of NAc terminals in the VTA prevents nicotine-induced inhibition of DA neurons by blocking inhibitory feedback.

### Inhibitory feedback from the NAc mediates nicotine-induced anxiety-like behavior

We next investigated the implication of this NAc-VTA inhibitory feedback in the anxiogenic effect of nicotine. Intraperitoneal (i.p) injection of nicotine (0.5 mg/kg) induced a decrease in the time spent in open arms in an elevated O maze (EOM), indicative of an anxiogenic effect of the drug (Fig. 5e), which was not correlated with locomotor activity over time (Fig. S6A). We have previously shown that this anxiogenic effect of nicotine requires the inhibition of VTA DA neurons projecting to the Amg[16]. To complete our demonstration of a functional VTA-NAc inhibitory loop, we examined whether blocking the inhibitory feedback originating from the NAc could alleviate the anxiogenic effect of nicotine. To this aim, we expressed JAWS or GFP in the three subareas of the NAc in WT mice and implanted optical fibers bilaterally in the VTA (Figs. 5f and S6B). Mice then received an i.p. injection of nicotine (0.5 mg/kg) 1 min before the EOM test and light stimulation in the VTA throughout the 9 min test. Photoinhibition of NAc terminals in the VTA during the EOM test abolished the anxiogenic effect of nicotine injection, as indicated by the percentage of time spent by JAWS-expressing mice in the EOM open arms that did not decrease during the test, in contrast to GFP-expressing mice (Fig. 5g). These effects were not correlated with locomotor activity over time (Fig. S6C). Without nicotine, photoinhibition of NAc-VTA terminals alone in JAWS-expressing mice had no effect on the distance traveled in the open field or the time spent in the open arms of the EOM test (Fig. S6D, E). Finally, considering that both nicotine and ethanol inhibit Amg-projecting DA neurons, we examined whether ethanol, like nicotine, could induce anxiety-like behavior through the NAc-VTA circuit. As observed with nicotine, inhibition of NAc terminals prevents the decrease in time spent in the open arms of the EOM over time following ethanol injection (Fig. 5h), without altering locomotor activity (Fig. S6F).

These data highlight the functional role of an inhibitory loop between the VTA and the NAc in mediating nicotine- and ethanol-induced anxiety-like behavior, primarily through the inhibition of DA neurons.

## Discussion

The notion that drugs reinforce behavior through a broad activation of the DA system is challenged by evidence showing distinct sub-populations of DA neurons in the VTA, each associated with specific appetitive, aversive, or attentional behaviors[7,8,26,27]. Our previous work has highlighted the heterogeneity of the VTA, particularly in response to nicotine[15,16]. Nicotine triggers both activation and inhibition of VTA DA neurons, associated with two distinct anatomical and functional circuits. These circuits produce distinct affective behaviors: inhibition of VTA DA neurons projecting to the BLA is anxiogenic[16,28], whereas activation of those projecting to the NAcLSh is rewarding[16]. During nicotine exposure, these dual DA pathways may act concurrently, as the same nicotine dose can be both rewarding and anxiogenic[16,29]. Here, we present evidence that inhibition and activation are not isolated events; rather, inhibition arises as a downstream consequence of activation and is mediated by feedback inhibition from the NAc. This finding has significant conceptual implications, suggesting that the VTA should not be viewed as a collection of independent subcircuits that function in isolation. Instead, it supports a more integrated and interconnected system, where activation in one pathway directly influences activity in another. We propose that nicotine-induced inhibition of DA neurons arises from two key properties of the VTA-NAc network, (1) its recurrent architecture and (2) the differential sensitivity of DA neurons projecting to the Amg or NAc to nicotine.

First, regarding the recurrent network architecture, recent studies have shown that VTA DA neurons form organized reciprocal connections with the NAc subregions they target, establishing distinct inhibitory feedback loops[23]. Our findings demonstrate that Amg-projecting DA neurons receive significant inhibitory inputs from all NAc subregions, and that optogenetic silencing of these inputs prevents nicotine-induced inhibition of VTA DA neurons. In support of this mechanism, we used a retrograde viral strategy to restore β2*nAChR expression specifically in NAc-projecting DA neurons in β2 knockout mice. This approach allows us to demonstrate that nicotine-induced activation of NAc-projecting neurons was sufficient to inhibit other DA neurons in the VTA, including those projecting to the Amg. This network-level mechanism is consistent with previous studies showing that intra-VTA infusion of nicotine at a rewarding dose increases the firing of NAc MSNs while inhibiting fast spiking interneurons[30]. Together these findings suggest that nicotine efficiently triggers inhibitory feedback from the NAc to the VTA via NAc-projecting DA neurons. However, our viral rescue strategy has limitations, such as the inability to specifically target DA neurons and potential variability in β2*nAChRs re-expression levels across neurons. In addition, other nAChR subtypes in the NAc or regions targeting the VTA may also be involved. Optogenetic activation of the VTA_{DA}-NAc pathway in DAT-Cre mice replicated the nicotine-induced activation/inhibition pattern, confirming that NAc-projecting DA neurons can trigger inhibition in VTA DA neurons. Optogenetic-induced inhibition of VTA neurons correlates with nicotine-induced inhibition, further suggesting that activation of the VTA_{DA}-NAc pathway is a key component of the DA inhibition observed with nicotine. Overall, these data indicate that activation of NAc-projecting DA neurons may trigger feedback inhibition in the VTA through NAc-VTA projections that particularly target Amg-projecting neurons.

Second, the differential sensitivity to nicotine observed between NAc- and Amg-projecting DA neurons likely reflects differences in nAChR composition or density, and may result in nicotine primarily activating NAc-projecting neurons, while Amg-projecting neurons are more prone to recurrent inhibition.[14,20] Amg-projecting neurons showed weaker nicotinic currents for the same nicotine dose, suggesting reduced nAChR sensitivity. Heterogeneity in nicotinic currents among DA neurons has been linked to differential expression of β2 or α7 containing nAChRs, with stronger currents linked to β2*nAChRs[31] that are likely expressed in both NAc- and Amg-projecting DA neurons. Therefore, the greater nicotine responsiveness of NAc-projecting neurons could potentially be due to the presence of specific subunit combinations such as α4β2*, α6β2*, or α4α6β2* nAChRs[32–34] or to specific subunit stoichiometries in these receptors. Another key candidate is the α5 subunit, which increases both expression levels and calcium permeability of α4β2*nAChRs, leading to stronger nicotine-evoked currents in slices and greater nicotine responses in vivo[35–37]. Importantly, desensitization rates also vary between nAChR subtype. For instance, α4α6β2 nAChRs desensitize more slowly than α4β2 nAChRs, allowing persistent activity of DA neurons in response to nicotine ex vivo[38]. It has been proposed that prolonged exposure to high doses of nicotine may desensitize nAChRs, thereby facilitating the suppression of NAc-projecting DA neuron activity by inhibitory inputs, ultimately leading to nicotine aversion[39]. However, in our study, it is unlikely that desensitization alone accounts for the inhibition of Amg-projecting neurons in vivo, nor for the lack of nicotine-evoked currents in these neurons ex vivo, starting from the lowest doses of acute nicotine.

We provide evidence that nicotine-induced activation of the VTA_{DA}-NAc pathway leads to inhibition of the VTA_{DA}-Amg pathway, highlighting a complex interplay between reward-related and emotion-related neural circuits during drug exposure. Nicotine exhibits both rewarding[3,14] and anxiogenic effects[40–42] that are linked to DA signaling. Inhibition of the VTA_{DA}-Amg pathway increases anxiety-like behavior, whereas its activation decreases it[16,28]. Conversely, activation of the VTA_{DA}-NAc pathway triggers online conditioned place preference, which is associated with reward signaling. Furthermore, counteracting the inhibition of BLA DA terminals during nicotine exposure abolishes nicotine-induced anxiety-like behavior, while

counteracting the activation of NAc lateral shell DA terminals reduces it[16]. Consistent with these findings, we abolished nicotine-induced anxiety-like behavior associated with inhibition of Amg-projecting DA neurons by inhibiting NAc terminals in the VTA. Therefore, NAc-VTA projections targeting Amg-projecting DA neurons may link the rewarding and negative emotional outcomes of nicotine intake. Recent studies have investigated the effect of optogenetic modulation of NAc-VTA terminals originating from discrete parts of NAc subregions in relation to anxiety-like behaviors. These studies show no effect on anxiety-like behavior with activation[23] and an anxiogenic-like effect with inhibition[43]. This may seem to contradict our findings showing that activation of NAc-VTA projections is necessary for nicotine-induced anxiety-like behavior. However, it is important to note that discrete optogenetic modulation, while effective for precise functional dissection of neural circuits, cannot recapitulate the simultaneous and widespread activation of networks that occurs during drug exposure. Indeed, nicotine indiscriminately activates DA neurons projecting to different subregions of the NAc[16], which in turn send inhibitory inputs to Amg-projecting DA neurons. Our approach of globally activating VTA$_{DA}$-NAc pathways or inhibiting all NAc-VTA projections was designed to mimic this broad circuit engagement. Under such conditions, our data support a model where nicotine-induced inhibition of Amg-projecting DA neurons - and the associated anxiety-like behavior - arises from convergent inhibitory inputs originating across the NAc. Nevertheless, we do not exclude the contribution of additional pathways to nicotine's anxiogenic effects, which may also account for differences between pharmacological and circuit-specific optogenetic manipulations[23,43].

The NAc and VTA share reciprocal connections that are anatomically and functionally specific[23,44]. While previous studies identified distinct VTA$_{DA}$-NAc subpopulations[23], connectivity between NAc subregions and Amg-projecting DA neurons have not yet been established. Here, we found that VTA$_{DA}$-BLA neurons receive GABAergic inputs from the NAcCore and preferentially from the NAcLSh over NAcMSh, distinguishing them from other medial VTA neurons that project to the NAcMSh[23]. Although these projections differ anatomically, BLA- and NAcMSh-projecting DA neurons have been involved in signaling salience rather than emotional valence[45–47]. According to our experiments, VTA$_{DA}$-BLA neurons are primarily targeted by inputs from the NAcCore, a subregion innervated by medial VTA[8] and known to encode salience[48]. These findings support a broader model of functional organization within the VTA, in which distinct DA circuits encode motivational salience versus valence.

Despite acting through different molecular mechanisms, nicotine and ethanol both increase DA release in the NAc[1]. A key finding of our study is that both drugs induce inhibition of Amg-projecting DA neurons and activation of NAc-projecting DA neurons. This shared effect suggests convergence onto a common inhibitory feedback loop, whereby activation of the VTA$_{DA}$-NAc pathway triggers NAc-VTA inhibition of Amg-projecting neurons. Supporting this idea, we found that inhibition of NAc-VTA projections disrupts ethanol's anxiogenic effects, suggesting that feedback inhibition from the NAc contributes to the impact of both substances on DA signaling. Recent findings indicate that ethanol produces greater activation of DA neurons in the lateral than in the medial VTA[17]. This suggests that, like nicotine, ethanol may override inhibitory mechanisms within a certain DA subpopulation. In contrast, natural rewards activate both DA pathways similarly[45,47], but with weaker and shorter-lasting responses, likely insufficient to trigger the NAc-VTA inhibitory loop. In addition, natural rewards may activate both NAc- and Amg-projecting neurons, overriding any feedback inhibition. This distinction suggests that selective engagement of the VTA-NAc feedback loop, resulting in opposing activity across reward and emotion-related circuits, may be a hallmark of drugs of abuse. Substances that act at the somatic level in the VTA (e.g., opioids, cannabinoids, benzodiazepines) may similarly trigger

this inhibitory feedback. Conversely, psychostimulants like cocaine and amphetamine, which increase DA release by acting directly on axon terminals, may bypass this loop entirely, potentially explaining their distinct emotional and behavioral profiles.

Finally, our findings offer a mechanistic basis for the high prevalence of nicotine and alcohol co-use[49–51]. The effects of both drugs are cumulative, activating NAc-projecting DA neurons via distinct though complementary mechanisms. Nicotine primarily acts through somatic nAChRs[3], whereas ethanol dampens GABA signaling in the VTA, leading to DA neuron disinhibition[52,53]. Critically, inhibition of Amg-projecting DA neurons appears to result from a shared, network-level process. This implies that both substances reinforce behavior by converging on DA release in the NAc, which could facilitate cross-substitution, enhance reward prediction error, or buffer withdrawal effects. In other words, stimulation of the reward system expected from one drug could be achieved through the use of the other, potentially explaining cross-sensitization. In the case of nicotine, it has been proposed that prolonged exposure leads to nAChR desensitization, reducing the overall rewarding properties of the drug, promoting DA neuron inhibition and an aversive state that may lead to cessation of intake[39]. Co-use with ethanol, which can activate DA neurons independently of nAChRs, may counteract nicotine-induced desensitization, sustaining DA neuron excitability and reinforcing nicotine's rewarding effects. Moreover, nicotine intake is regulated by a negative feedback mechanism involving the medial habenula–interpeduncular nucleus (MHb-IPN) and lateral habenula–rostromedial tegmental nucleus (LHb-RMTg) pathways, which ultimately inhibit DA neurons[54,55]. Therefore, the reduction of VTA GABA signaling by ethanol may interfere with these inhibitory pathways, promoting excessive nicotine intake. This mechanism could underlie the common observation that, in humans, tobacco consumption increases during drinking episodes[56].

Our study supports the idea that the rewarding effects of nicotine are intrinsically linked to the emergence of negative emotional states. We demonstrate that acute nicotine induces anxiety-like behavior as a direct consequence of activating the DA reward pathway that underlies reinforcement. Thus, anxiogenic and rewarding signals not only occur simultaneously upon nicotine exposure but are also functionally intertwined. This interdependence implies that alterations in the VTA$_{DA}$-NAc pathway, whether pharmacological or pathological, may change nicotine's emotional impact through its effects on the VTA$_{DA}$-Amg pathway. It also suggests that the circuits underlying negative emotional states, often referred to as the anti-reward system[57], and those mediating reward evolve in tandem during chronic nicotine exposure and potentially during withdrawal. This raises intriguing questions about how these two circuits with opposing messages compete to shape nicotine reinforcement, and whether an imbalance between them could favor the transition to substance use disorders. Our findings highlight the neural circuits architecture through which nicotine exerts its dual affective effects, reinforcing the view that addiction is not solely a disorder of reward motivation, but also of emotional regulation. Understanding these interactions could pave the way for more effective treatments for nicotine addiction and other substance use disorders where both reward and emotional regulation are involved.

## Methods

### Animals
Experiments were performed on wild-type (WT) C57Bl/6Rj (Janvier Labs, France), DATiCRE (DAT-Cre), DAT-D2R$^{-/-}$, ACNB2$^{-/-}$ (β2$^{-/-}$) mice, weighing 25–35 grams. β2$^{-/-}$ mice were provided by the team of Uwe Maskos (Pasteur institute, Paris, France). DATiCRE mice were provided by François Tronche's team (IBPS Paris, France), and were bred on site and genotyped as described[58]. DAT-D2R$^{-/-}$ were provided by Emmanuel Valjent's team (IGF, Montpellier, France). WT and DATiCRE (DAT-Cre)

animals used for in vivo recordings and behavioral experiments were males. The decision to use only one sex in WT mice was based on the nature of the experiment, which required juxtacellular labeling of recorded neurons identified by retrograde tracing. This method yields a low number of double-labeled neurons per animal when using retrobeads and neurobiotin, so a large number of animals is required to obtain sufficient data. Splitting the data by sex would make post hoc statistical comparisons between sexes impractical unless an even larger cohort was used. The use of a single sex in DAT-Cre mice for in vivo recordings combined with optogenetics, which involves a limited number of recordings per animal, was justified for the same reason. Finally, we chose to establish the correlation between electrophysiology and behavior by testing mice of the same sex in both paradigms. DAT-D2R$^{-/-}$ and ACNB2$^{-/-}$ transgenic animals used for in vivo electrophysiology were equally divided between males and females to optimize animal yield. The response of DA neurons to nicotine was similar in both sexes (Fig. S7).

Mice were kept in an animal facility where temperature ($20 \pm 2\,°C$) and humidity were automatically monitored and a circadian 12/12-h light-dark cycle was maintained. All experiments were performed on 8–20-week-old mice.

All experiments were performed in accordance with the recommendations for animal experiments issued by the European Commission directives 219/1990, 220/1990 and 2010/63, and approved by Sorbonne University and PSL.

## Virus

Lentiviruses were prepared in Pasteur institute as previously described[3,14], with a titer of either 260 ng/µl for the AChR β2-expressing vector (PDGF.low.mCherry.lox.β2) or 370 ng/µl for GFP-expressing vector (PDGF.low.mCherry.lox.β2). pAAV5-hSyn-hChR2(H134R)-EYFP and pAAV-hsyn-Jaws-KGC-GFP-ER2 were provided by Addgene. ssAAV-retro/2-hSyn1-chI-mCherry_2A_iCre-WPRE-SV40p(A), ssAAV-retro/2-EYFP(rev)-dlox-WPRE-hGHp(A), ssAAV5/2-hSyn1-JAWS-KGC-EGFP-ER2-WPRE-hGHp(A), ssAAV5/2-hSyn1-EGFP- WPRE-hGHp(A) were provided by VVF Zurich.

## Drugs

In all our experiments we used a nicotine hydrogen tartrate salt (Sigma-Aldrich, USA) and liquid ethanol 96% (Emprove, Ph Eur BP, Sigma-Aldrich).

For juxtacellular and tetrode recordings, we performed an intravenous injection (i.v) of nicotine at a dose of 30 µg/kg or ethanol at a dose of 250 mg/kg (and 125, 500, 750 for dose-response experiments) or saline solution ($H_2O$ with 0.9% NaCl). For patch-clamp recordings, we used 30 µM of nicotine for bath-application and 10, 30 or 100 µM for local puffs. All solutions were prepared in the laboratory.

## Stereotaxic surgeries

For viral or retrobead injections, mice were anesthetized with a gas mixture of oxygen (1 L/min) and 3% isoflurane (IsoFlo) through a TeamSega apparatus. Mice deeply anesthetized were then placed in a stereotaxic frame (David Kopf), maintained under anesthesia throughout the surgery at 3–2% isoflurane. The skin was shaved, disinfected and locally anesthetized with 100 µl of lurocaine 10% at the location of the scalp incision. Depending on the experiments, unilateral or bilateral craniotomies were then performed over the VTA, NAc or BLA (see details below). At the end of the surgery, 75 µl of buprenorphine (Buprecare, 0.1 mg/kg) was injected subcutaneously to prepare awakening.

## Retrobead injections

Green or red fluorescent retrograde tracers, retrobeads (RB, Lumafluor), were injected using a cannula (diameter 36G, Phymep, Paris, France). Red RB were used for optogenetic experiments in slice electrophysiology, to not overlap with the YFP signal. Green RB were used for juxtacellular recordings and for patch-clamp recordings without optogenetics. The canula was connected to a 10 µl Hamilton syringe (Mode 1701, Hamilton Robotics, Bonaduz, Switzerland) placed in a pump (QSI, Stoelting Co, Chicago, IL, USA). For juxtacellular recordings, injections were performed in the 3 subregions of the NAc (NAc medial shell NAcMSh: bregma 1.78 mm, lateral 0.45 mm, ventral 4.1 mm; NAc core: bregma 1.55 mm, lateral 1.0 mm, ventral 4.0 mm; NAc lateral shell NAcLSh: bregma 1.45 mm, lateral 1.75 mm, ventral 4.0 mm) or in Amg (BLA: bregma −1.35 mm, lateral 3.07 mm, ventral 4.7 mm; CeA: bregma − 0.78 mm, lateral 2.3 mm, ventral 4.8 mm). For patch-clamp experiments, only NAcLSh (bregma 1.45 mm, lateral 1.75 mm, ventral 4.0 mm) and BLA (bregma −1.35 mm, lateral 3.07 mm, ventral 4.7 mm) were targeted with the same protocol of RB injection. To enable retrograde transport of the RB into the somas of midbrain DA neurons, we waited 2 weeks after injection into the NAc and 3–4 weeks after injection into the Amg, before running electrophysiology experiments.

## Lentiviral reexpression and optogenetic experiments

All viral injections were done using a glass micropipette (10 µl graduated borosilicate glass capillary; Wiretrol I Calibrated Micropipettes, Drummond) prefilled with mineral oil, and fixed into the MO-10 One-axis Oil Hydraulic Micromanipulator (Narishige).

For patch-clamp experiments, activation of NAc terminals was mediated through viral injection of pAAV5-hSyn-hChR2(H134R)-EYFP ($2 \times 10^{13}$, 26973, Addgene).

To perform re-expression of the β2 subunit specifically in the VTA-NAc pathway of β2$^{-/-}$ mice, we first injected ssAAV-retro/2-hSyn1-chI-mCherry_2A_iCre-WPRE-SV40p(A) ($8.4 \times 10^{12}$, v147-retro, VVF Zurich) in the 3 subregions of the NAc (200 nl in each site: NAcMSh: bregma 1.78 mm, lateral 0.45 mm, ventral 4.1 mm; NAcCore: bregma 1.54 mm, lateral 1.0 mm, ventral 4.0 mm; NAcLSh: bregma 1.45 mm, lateral 1.75 mm, ventral 4.0 mm) and we waited 3 weeks for Cre recombinase to expressed. We then performed unilateral injections of 700 nl of PDGF.lox.mCherry.lox.GFP (185 ng/nl, Pasteur institute) or PDGF.lox.mCherry.lox.β2 (260 ng/nl, Pasteur institute) in the VTA (bregma −3.1 mm, lateral 0.5 mm, ventral 4.5 mm).

To perform DA-neuron specific optogenetic activation of the VTA-NAc pathway for patch-clamp or tetrode recordings, we used 8-week-old DAT-Cre mice, in which Cre recombinase expression is restricted to DA neurons without disrupting endogenous dopamine transporter (DAT) expression[58,59]. We injected ssAAV-retro/2-hEF1a-dlox-hChR2(H134R)_EYFP(rev)-dlox-WPRE-hGHp(A) ($5.4 \times 10^{12}$, v214-retro, VVF Zurich) in the 3 subregions of the NAc (200 nl each site: NAcMSh: bregma 1.78 mm, lateral 0.45 mm, ventral 4.1 mm; NAcCore: bregma 1.55 mm, lateral 1.0 mm, ventral 4.0 mm; NAcLSh: bregma 1.45 mm, lateral 1.75 mm, ventral 4.0 mm) and we waited 3 weeks before recordings.

To perform non-conditional optogenetic inhibition of the NAc inputs onto the VTA for tetrode recordings, we injected pAAV-hsyn-Jaws-KGC-GFP-ER2 ($1,3e^{13}$ 1:10 dilution; 65014; Addgene) or ssAAV5/2-hSyn1-JAWS-KGC-EGFP-ER2-WPRE-hGHp(A) ($5,9e^{12}$; v387-5; VVF Zurich, from #65014 of Addgene) in the 3 subregions of the NAc (200 nl each site: NAcMSh: bregma +1.78 mm, lateral 0.45 mm, ventral 4.1 mm; NAcCore: bregma +1.54 mm, lateral 1.0 mm, ventral 4.0 mm; NAcLSh: bregma +1.45 mm, lateral 1.75 mm, ventral 4.0 mm) in WT mice.

To perform optogenetic behavioral experiments (EOM test, Open field), we did bilateral injections of ssAAV5/2-hSyn1-JAWS-KGC-EGFP-ER2-WPRE-hGHp(A) ($5,9e^{12}$; v387-5; VVF Zurich, from #65014 of Addgene) or ssAAV5/2-hSyn1-EGFP- WPRE-hGHp(A) ($5,7e^{12}$; v81-5; VVF Zurich) in the NAc (200 nl each site: NAcMSh: bregma +1.78 mm, lateral 0.45 mm, ventral 4.1 mm; NAcCore: bregma +1.54 mm, lateral 1.0 mm, ventral 4.0 mm; NAcLSh: bregma +1.45 mm, lateral 1.75 mm, ventral 4.0 mm) in WT mice. Optical fibers (200 mm core, NA = 0.39,

ThorLabs) coupled to a zirconia ferule (1.25 mm) were implanted bilaterally in the VTA (10° angle; bregma −3.07 mm, lateral 1.43 mm, ventral −3.9 mm) and fixed to the skull with dental cement (SuperBond, Sun medical).

### In vivo electrophysiology on anesthetized animals

Induction of anesthesia was done with gas mixture of oxygen (1 L/min) and 3% isoflurane (IsoFlo) through a TeamSega apparatus. Mice deeply anesthetized were then placed in a stereotaxic frame (David Kopf), maintained under anesthesia throughout the surgery at 3–2.5% isoflurane. The scalp was opened and a cranial window was drilled in the skull above the location of the VTA (coordinates: 3.1 ± 3 mm posterior to bregma, 0.4 to 0.5 mm lateral to the midline, 3.9 to 5 mm ventral from the brain). During recordings, mice were maintained deeply anesthetized at 2 % isoflurane, with monitoring and adjustment of the anesthesia throughout the experiment. Intravenous (i.v.) administration of saline, nicotine (30 µg/kg) or ethanol (250 mg/kg) was carried out through a catheter (30G needle connected to polyethylene tubing PE10) connected to a Hamilton syringe, into the saphenous vein of the animal. For multiple doses of ethanol, mice received two to four injections of 125, 250, 500, and/or 750 mg/kg (pseudo-randomly administrated). For experiments comparing the effects of single and co-injection of nicotine and ethanol, animals received either 30 µg/kg of nicotine and 500 mg/kg of ethanol simultaneously or separately, with administration occurring in a pseudo-randomized manner.

**Juxtacellular recordings.** To perform single unit extracellular recordings, recording electrodes were pulled from borosilicate glass capillaries (Harvard Apparatus, with outer and inner diameters of 1.50 and 1.17 mm, respectively) with a Narishige electrode puller. The tips were broken under microscope control and filled with 0.5% sodium acetate containing 1.5% of neurobiotin tracer (VECTOR laboratories). Electrodes had tip diameters of 1–2 µm and impedances of 6–9 MΩ. A reference electrode was placed in the subcutaneous tissue. The recording electrodes were lowered vertically through the hole with a micro drive. Electrical signals were amplified by a high-impedance amplifier (Axon Instruments) and monitored through an audio monitor (A.M. Systems Inc.). The unit activity was digitized at 25 kHz and recorded using Spike2 software (Cambridge Electronic Design) for later analysis. Individual electrode tracks were separated from one another by at least 0.1 mm in the medio-lateral axis. The electrophysiological characteristics of dopamine neurons were analyzed in the active cells encountered when passing the microelectrode in a stereotaxically defined block of brain tissue corresponding to the coordinates of the VTA (coordinates: between 3 and 3.4 mm posterior to bregma, 0.4 to 0.6 mm lateral to midline, and 3.9 to 5 mm below brain surface). Extracellular identification of dopamine neurons was based on their location as well as on the set of unique electrophysiological properties that distinguish dopamine from non-dopamine neurons in vivo: (i) a typical triphasic action potential with a marked negative deflection; (ii) a long duration (>2.0 ms); (iii) an action potential width from start to negative trough >1.1 ms; (iv) a slow firing rate (<10 Hz and >1 Hz). After recording, nicotine-responsive cells were labeled by electroporation of their membrane: successive currents squares were applied until the membrane breakage, to fill cell soma with neurobiotin contained into the glass pipet[60]. To be able to establish correspondence between neurons responses and their localization in the VTA, we labeled one type of response per mouse: solely activated neurons or solely inhibited neurons, with a limited number of cells per brain (1–4 neurons maximum, 2 by hemisphere), always with the same concern of localization of neurons in the VTA. All data were analyzed with R.

**Multi-unit extracellular recordings.** 4–5 weeks after viral infection, we used a Mini-Matrix (Thomas Recording) allowing us to lower within the VTA up to 3 tetrodes (Tip shape A, Thomas Recording, $Z = 1–2$ MW) and a tip-shaped quartz optical fiber (100 mm core, NA = 0.22, Thomas Recording) for photostimulation. The fiber was coupled to a dual LED (450–465 nm for ChR2, 600–630 nm for Jaws, Prizmatix) with an output intensity of 200–500 mW for both wavelengths. These four elements could be moved independently with micrometer precision. For ChR2-mediated activation of NAc-projecting neurons, tetrodes were lowered in the medial part of the VTA (coordinates: 0.3–0.4 mm lateral to midline) while the optical fiber was lowered in the lateral part of the VTA (coordinates: 0.6–0.7 mm lateral to midline) and 100 µm above the tetrodes (around 300 µm between the fiber and the tetrodes). Photostimulation was delivered in a regular pattern at 10 Hz (pulse of 5 ms duration delivered every 100 ms for 5 min) or in a burst pattern at 20 Hz (a sequence of five 5 ms pulses separated by 50 ms delivered every 500 ms for a total duration of 5 min). For JAWS mediated inhibition of NAc-VTA terminals tetrodes and optical fiber were lowered in the medial part of the VTA (coordinates: 0.3–0.4 mm lateral to midline, around 100 µm between the fiber and the tetrodes). Photostimulation was delivered continuously for 5 min.

Spontaneously active putative DA neurons were identified on the basis of the same electrophysiological criteria used for juxtacellular recordings. Baseline activity was recorded for 5 min, prior to i.v. injection of nicotine, allowing us to target drug-inhibited neurons and a second i.v. injection of drug with optical-stimulation was then performed.

Electrophysiological signals were acquired with a 20 channels preamplifier included in the Mini Matrix (Thomas Recording) connected to an amplifier (Digital Lynx SX 32 channels, Neuralynx) digitized and recorded using Cheetah software (Neuralynx). Spikes were detected (CSC Spike Extractor software, Neuralynx) and sorted using a classical principal component analysis associated with a cluster cutting method (SpikeSort3D Software, Neuralynx). All the data were analyzed with R.

### Ex vivo electrophysiology: patch clamp recordings

Mice were deeply anesthetized by an intraperitoneal injection of a mix of ketamine (150 mg/kg Virbac 1000) and xylazine (60 mg/kg, Rompun 2%, Elanco). Coronal midbrain sections (250 µm) were sliced with a Compresstome (VF-200, Precisionary Instruments) after intracardial perfusion of cold (4 °C) sucrose-based artificial cerebrospinal fluid containing (in mM): 125 NaCl, 2.5 KCl, 1.25 $NaH_2PO_4$, 26 $NaHCO_3$, 5,9 $MgCl_2$, 25 sucrose, 2.5 glucose, 1 kynurenate (pH 7.2, 325 mOsm). After 8 min at 37 °C for recovery, slices were transferred into oxygenated artificial cerebrospinal fluid (ACSF) containing (in mM): 125 NaCl, 2.5 KCl, 1.25 $NaH_2PO_4$, 26 $NaHCO_3$, 2 $CaCl2$, 1 $MgCl2$, 15 sucrose, 10 glucose (pH 7.2, 325 mOsm) at room temperature for the rest of the day. Slices were individually transferred to a recording chamber continuously perfused at 2 mL/min with oxygenated ACSF. Patch pipettes (4–6 MΩ) were pulled from thin wall borosilicate glass (G150TF-3, Warner Instruments) with a micropipette puller (P-87, Sutter Instruments Co.). Neurons were visualized using an upright microscope coupled with a Dodt gradient contrast imaging, and illuminated with a white light source (Scientifica). Whole-cell recordings were performed with a patch-clamp amplifier (Axoclamp 200B, Molecular Devices) connected to a Digidata (1550 LowNoise acquisition system, Molecular Devices). Signals were low-pass filtered (Bessel, 2 kHz) and collected at 10 kHz using the data acquisition software pClamp 10.5 (Molecular Devices). VTA location was identified under microscope. Identification of dopaminergic neurons was performed by location and by their electrophysiological properties (width and shape of action potential (AP) and after hyperpolarization (AHP)).

To perform recordings of spontaneous inhibitory post-synaptic currents (sIPSCs), we used a Cesium-based internal solution of (in mM): 130 CsCl, 1 EGTA, 10 HEPES, 2 MgATP, 0.2 NaGTP, 0.1% neurobiotin pH 7.35 (270-285 mOsm). Local perfusion was used to apply nicotine locally in the bath above the recorded cells. Recordings of light-evoked GABA current from activation of NAc terminals were conducted in the presence of 20 μM DNQX (6,7-Dinitroquinoxaline-2,3-dione, HelloBio) and 50 μM D-AP5 (D-(-)−2-Amino-5-phosphono-pentanoic acid, HelloBio) to block respectively AMPA and NMDA receptors, 500 mM of TTX (tetrodotoxin citrate, HelloBio) to block voltage-gated sodium channels and 1 mM of 4-AP (4-aminopyridine, HelloBio) to block voltage-gated potassium channels[61]. Light-evoked responses were obtained every 10 s with one pulse (3 ms) of 460 nm wavelength (5 sweeps for each neuron). For recordings of miniature inhibitory post-synaptic currents (mIPSCs), Cs-based internal solution was used with 20 μM DNQX, 50 μM D-AP5 and 500 nM TTX.

To perform characterization of DA subpopulations and puffs recordings (200 ms, 2 psi), potassium gluconate-based intracellular solution was used containing (in mM): 135 K-gluconate, 10 HEPES, 0.1 EGTA, 5 KCl, 2 MgCl$_2$, 2 ATP-Mg, 0.2 GTP-Na, and biocytin 2 mg/mL (pH adjusted to 7.2). The same internal solution was used to record optical stimulation of retroChR2 virus in NAc-projecting DA neurons. To characterize retroChR2 expression, 10 and 20 Hz (3 ms pulse; train rate of 10 or 20 Hz; 5 sweeps of 10 s) photostimulation were used to drive neuronal firing in current-clamp mode. LED (Prizmatix) intensity was set to 30%, as no higher amplitude currents were observed above this threshold.

For mIPSCs and sIPSCs, we used Clampfit (Molecular Devices) to extract individual currents recorded for at least 5 min, based on a template search made of at least 15 example currents for each neuron. We kept currents of at least three times the noise standard deviation. We used Campfit to detect currents after puff application of nicotine or optogenetic stimulation, by comparing the peak amplitude with the baseline region.

### Behavioral task

**Elevated O maze test**. The raw data for behavioral experiments were acquired as video files. The elevated O-maze (EOM) apparatus consists of two open (stressful) and two enclosed (protecting) elevated arms that together form a zero or circle (diameter of 50 cm, height of 58 cm, 10 cm-wide circular platform). Time spent in exploring enclosed versus open arms indicates the anxiety level of the animal. The first EOM experiment assessed the effect of an i.p. injection of Nic (0.5 mg/kg) on WT mice. The test lasts 10 min: mice are injected 1 min before the test, and then put in the EOM for 9 min. In the second EOM experiment, i.p. injection of Nic (0.5 mg/kg) or Ethanol (1 g/kg) was combined with continuous optogenetic inhibition of NAc terminals targeting the VTA. Finally, we did an optogenetic EOM experiment to control for basal anxiety-like behavior under light-stimulation, during 15 min, alternating 5 min-periods of stimulation and non-stimulation (OFF-ON-OFF). Time spent in open or closed arms was extracted frame-by frame using the open-source video analysis pipeline ezTrack[62]. Mice were habituated to the stress of handling and injection for a minimum of 1 week before testing.

**Open field test**. The open field (OF) is a square enclosure of 50 cm × 50 cm where animals can move freely. Distance traveled by animals was quantified over time. Regarding the optogenetic experiment conducted in the OF, animals were placed in the maze for 15 min, while alternating between OFF, ON and OFF optical stimulation periods of 5 min each.

### Fluorescence immunohistochemistry

Recorded neurons in patch-clamp experiments were filled with biocytin in order to validate the presence of TH enzyme by immunohistochemistry, indicator of dopaminergic neurons. After recordings, slices were fixed in 4% PFA (paraformaldehyde) overnight. Recorded neurons in juxtacellular expriments were filled with neurobiotin and after euthanasia of the animals, brains were rapidly removed and fixed in 4% PFA. After a period of at least 3 days of fixation at 4 °C, serial 60-μm sections were cut from the midbrain with a vibratome.

Immunostaining experiments were performed as follows: free-floating VTA brain sections were incubated for 3 h (juxtacellular experiments) or 6 h (patch-clamp experiments) at 4 °C in a blocking solution of phosphate-buffered saline (PBS) containing 3% bovine serum albumin (BSA, Sigma; A4503) and 0.2% Triton X-100 (vol/vol), and then incubated overnight (juxtacellular experiments) or during 72 h (patch-clamp experiments) at 4 °C with a mouse anti-tyrosine hydroxylase antibody (anti-TH, Sigma, T1299), at 1:500 or 1:200 dilution, in PBS containing 1.5% BSA and 0.2% Triton X-100. Sections were then rinsed with PBS, and incubated for 3 h (juxtacellular experiments) or 6 h (patch-clamp experiments) at room temperature with Cy3-conjugated anti-mouse and AMCA-conjugated streptavidin (Jackson ImmunoResearch) both at 1:500 or 1:200 dilution in a solution of 1.5% BSA in PBS. After three rinses in PBS, slices were wet-mounted using Prolong Gold Antifade Reagent (Invitrogen, P36930). In the case of optogenetic or re-expression experiments, identification of trans-fected neurons by immunofluorescence was performed as described above, with addition of chicken anti-GFP primary antibody (1:500, ab13970, Abcam) in the 3% BSA solution. A goat-anti-chicken Alexa-Fluor 488 (1:500, Life Technologies) was then used as secondary antibody.

Microscopy was performed using an epifluorescent microscope (Zeiss, AXIO Imager.M1), and images captured using a camera and analyzed with ImageJ. Immunoreactivity for both TH and biocytin or neurobiotin allowed us to confirm the neurochemical phenotype of DA neurons in the VTA or the transfection success.

### Quantification and analysis of in vivo electrophysiological recordings

Subpopulations of DA neurons were automatically classified using variation of firing frequency induced by nicotine or ethanol i.v injection. First, we calculated the maximal variation from the baseline per neuron, within the first 3 min following injection. We then used a bootstrapping method to exclude non-responding neurons.

For each neuron, the maximal and the minimal value of firing frequency was measured within the response period (3 min) that followed nicotine or saline injection, and the biggest of the two were used to determine the maximum of variation induce by the given injection. The effect of nicotine or alcohol was assessed by comparing the maximum of firing frequency variation induced by the drug and by saline injection. If the distribution of all maximum variation induced by nicotine or ethanol were different from those induced by saline, each neuron was then individually classified as activated or inhibited by bootstrapping. Baseline spike intervals were randomly shuffled 1000 times. Firing frequency was estimated on 60s-time windows, with 15 s time steps. For each neuron, we determined the percentile from the shuffled data corresponding to the drug-evoked response (max or min frequency after nicotine injection). Neurons were individually considered as responsive to nicotine or ethanol injection if this percentile is ≥0.95 or ≤0.05. Responsive neurons displaying an increase in firing frequency (Δf > 0) were defined as "Nic+" or "EtOH+" while neurons displaying a decrease in firing frequency (Δf < 0) were defined as "Nic−" or "EtOH−". For the dose-response curve, neurons were classified as EtOH+ or EtOH- based on their response for the first doses inducing a significant firing variation identified by bootstrapping.

The mean responses to nicotine and ethanol of activated and inhibited neurons are thus presented as a percentage of variation from baseline (mean $n$ ± SEM). Then, for activated (or inhibited) neurons, we compare the maximum (or minimum) value of firing frequency before

and after injection for nicotine, ethanol, and saline. Finally, to ensure that the drug-induced responses were different from the activation or inhibition induced by saline injection, we compared the firing variation induced by nicotine or ethanol with that induced by saline

To map the locations of EtOH+ and EtOH- neurons on an atlas, we included all neurons identified as ethanol-responsive by bootstrapping, regardless of whether they received nicotine or saline injections.

To examine the correlation between ethanol- and nicotine-induced responses, we included all neurons identified as ethanol-responsive by bootstrapping that received a nicotine injection, regardless of whether they received saline injection or not.

### Statistical analysis

All statistical analyses were done using the R software with home-made routines. Results were plotted as mean ± SEM. The total number (n) of observations in each group and the statistical tests used for comparisons between groups or within groups are indicated directly on the figures or in the "results" sections. Comparisons between means were performed with parametric tests such as Student's t-test, or two-way ANOVA for comparing two groups when parameters followed a normal distribution (Shapiro-Wilk normality test with $p > 0.05$), or Wilcoxon non-parametric test if the distribution was skewed. Holm's sequential Bonferroni post hoc analysis was applied, when necessary. Statistical significance was set at $p < 0.05$ (*), $p < 0.01$ (**), or $p < 0.001$ (***), or $p > 0.05$ was considered not to be statistically significant. For the surrogate analyses, nicotine and ethanol responses from the experimental data groups were pooled to reconstruct a new surrogate data set. The percentage of correlation of drug responses from the surrogate was then calculated (number of pairs of responses showing the same polarity for the two drugs). We plotted the density of the percentage correlation from 10,000 surrogates and counted the number of times this percentage reached the level of the percentage of correlation observed with experimental data.

### Statistics and reproducibility

All experiments were replicated with success.

### Reporting summary

Further information on research design is available in the Nature Portfolio Reporting Summary linked to this article.

## Data availability

Source data are provided with this paper. In addition, the data generated in this study have been deposited in the Zenodo database at https://doi.org/10.5281/zenodo.15517422. Source data are provided with this paper.

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

## Acknowledgements

We are grateful to Otilia de Oliveira and Emilie Tubeuf for animal caretaking support. This work was supported by Centre National de la Recherche Scientifique (CNRS; UMR8249), Agence Nationale de la Recherche (Vampire ANR-19-CE16-0001 and Stratagem ANR-23-CE17-0053-03 to F.M.), the French National Cancer Institute Grant (TABAC-16-01, TABAC-19-020 and SPA-21-002 to P.F.), the Foundation for Medical Research (FRM, Equipe FRM DEQ2013326488 to P.F.), the Labex memolife (to J.J. and E.V.), Swedish Research Council (2022-06168 to N.G.), NIDA–Inserm Postdoctoral Drug Abuse Research Fellowship (to L.M.R). Our team is affiliated with PSL-NEURO, funded by PSL University, and to *DIM C-BRAINS, funded by the Conseil Régional d'Ile-de-France.*

## Author contributions

T.L.B., P.F., and F.M. designed the study. T.L.B., P.F., and F.M. analyzed the data. T.L.B., C.N., and F.M. performed and analyzed in vivo juxtacellular electrophysiological recordings. T.L.B. designed, performed and analyzed ex vivo patch clamp recordings. E.V. contributed to ex vivo patch clamp recordings. T.L.B. performed stereotaxic injections (with contribution from C.S.), fiber implantations and behavioral optogenetic experiments (with contribution from E.V. and L.D.). T.L.B., E.V., L.D., A.G. performed PFA mice intracardiac perfusion and immunostaining experiments. J.J. and N.G. performed recordings and signal treatment for in vivo multi-unit experiments. S.T. contributed to in vivo juxtacellular electrophysiological recordings. S.M. contributed to setting up ex vivo patch clamp recordings. C.S., L.M.R., and A.M. contributed to design behavioral optogenetic experiments. S.P. and U.M. provided viruses.

E.V.J. provided DAT-D2-knockout (KO) mice. U.M. provided ACNB2-knockout (KO) mice. T.L.B., P.F., and F.M. wrote the manuscript.

## Competing interests

The authors declare no competing interests.
