## [Transparent Peer Review file · Nature Communications]

Nicotine engages a VTA-NAc feedback loop to inhibit amygdala-projecting dopamine neurons and induce anxiety-like behaviors.

Corresponding Author: Dr Fabio Marti

Version 0:

Reviewer comments:

Reviewer #1

(Remarks to the Author)

In their manuscript, Le Borgne et al. employ a broad range of cutting-edge techniques to compellingly reveal an inhibitory feedback circuit in ventral tegmental area (VTA) dopaminergic neurons projecting to the amygdala (DA VTA→Amygd), which induces anxiety following nicotine or alcohol intake. This circuit is activated via dopaminergic neurons projecting to the nucleus accumbens (DA VTA→NAc). Using in vivo juxtacellular single-neuron recordings combined with retrograde labeling, they demonstrate that both nicotine and alcohol induce activation or inhibition of DA neurons depending on their projection target—activation for neurons projecting to the NAc and inhibition for those projecting to the amygdala.

To better understand the inhibition of DA VTA→Amygd neurons, the authors employed patch-clamp techniques, showing that nicotine preferentially activates DA VTA→NAc neurons over DA VTA→Amygd neurons, independent of GABAergic interneurons in the VTA. Further, through selective rescuing of the beta2 nAChR's subunit in either VTA→NAc or VTA→Amygd neurons in beta2KO mice, as well as in DAT-Drd2 cKO mice, they excluded the involvement of intra-VTA GABAergic or intra-VTA dopaminergic circuits in nicotine's inhibitory effects.

By combining optogenetics with patch-clamp recordings, Le Borgne et al. demonstrated that a GABAergic projection from the NAc to the VTA inhibits DA VTA→Amygd neurons. This pathway, activated by DA VTA→NAc neurons, was confirmed using optogenetic approaches alongside in vivo electrophysiological recordings. Finally, using loss-of-function methods with inhibitory opsins, the authors confirmed the necessity of this pathway for nicotine's inhibitory effect on VTA DA neurons and for the anxiogenic effects induced by both nicotine and alcohol.

Together, these findings offer significant insights into the complex neural circuits involving VTA dopaminergic neurons, particularly in the context of addiction to nicotine and alcohol. Overall, this is a timely and high-quality study that is poised to make a substantial impact on the field of addiction neuroscience and the physiological understanding of VTA dopaminergic neurons. I have a few comments and suggestions regarding the work.

Abstract:

Line 26: The 'it' is kind of ambiguous. I guess it refers to nicotine

Results:

A critical finding from the paper is that most of the recorded DA neurons respond with the same polarity to nicotine and ethanol. This finding is only illustrated in Fig S1D-E. I think it should appear in the main figure.

While most recorded cells respond to nicotine and ethanol with the same polarity, about 20% of them are nicotine-inhibited and ethanol-activated and none are found to be nicotine-activated and ethanol-inhibited. I think this finding should be highlighted and discussed especially as they represent roughly a similar fraction than nicotine-inhibited ethanol-inhibited neurons.

I think the author's claim related to the similarity in medio-lateral segregation between ethanol and nicotine responses would

be reinforced if they directly compare the medio-lateral distribution of nicotine-activated neurons and ethanol-activated neurons. Same thing with nicotine-inhibited neurons and ethanol-inhibited neurons.

In lines 107-111, it is not explicit it refers to nicotine injections.

Double labelling in Figure S2B and D is hard to see. Maybe add arrows to point double-fluorescence, split channels, or add zoom
Scale bars are missing in Figure S2B to D

In line 130, the authors suggest a 'differential expression of nAChRs expression'. Is there any experimental evidence supporting this?

In lines 135-140, I'm not sure I understand the authors' rationale to explain the increase in the amplitude of sIPSC in NAc-projecting neurons.

The design of Figure 3C feels confusion. Maybe an alternative could be to split it into pie charts for % connectivity and bar graphs for the current amplitude.

The difference in the proportion of VTA=>Amygd neurons responding to stimulation of NAc=>VTA neurons according to their location in the NAc (Fig 3C) is not discussed.

YFP labelling and co-expression with TH is hard to see in Figure 3D right panel. Also the scale bar is missing.

The authors should rename what they call 10 or 20 Hz optogenetic stimulation consistently throughout the text to make it clear that one is constant and the other is pulsed.

Regarding optogenetic stimulations and recordings of DA neurons, I don't understand why excited ChR2-transfected DA neurons don't reach average frequencies close to the light stimulation patterns.

Did the authors try some opto-tagging experiments to precisely separate ChR2-expressing neurons from the rest of DA neurons?

In Figure S5E, the effect observed after Opto+Nic looks higher in magnitude than the effect of Nic alone. Is this true? Also this figure shows a significant effect of saline injection. This should be discussed.

In Figure S6 the authors should also display the effects on general locomotion of ethanol and optogenetic inhibition in the EOM (like figure S6C for nicotine)

(Minor)

For the consistency between panels, please change the order of pie charts in Fig 1L.

Figure S2D could be modified to also separate Nic-/EtOH+ neurons.

The distribution histograms in Fig S1F and S2E should be revised as the overlay between bars is unclear.

Horizontal axis labels are missing in left panels in Fig S1F and S2F

In Figures 2A-C, and S3, please keep the same order between NAc-DA and BLA-DA neurons.

Line 191: n = 8/13 and not n = 8/1.

Discussion:

Line 276: I'm not sure I would call anxiety and reward 'opposite behaviors'

Lines 305 to 311- of the discussion is highly (too?) speculative.

Indeed:

Lines 305-307: There is no evidence that the abundance of the beta2 subunit varies according to the particular type of DA neuron. The hypothesis of different combinations of beta2*nAChR receptors is an undiscussed alternative.

Lines 307-309: This sentence is not clear. As rescuing is specific to VTA=>NAc neurons, other VTA neurons and in particular VTA=>Amygdala neurons do not have beta2 de facto. Or are you implying that in a control mouse the VTA-Amyd neurons would have no beta2?

Lines 309-311: In reference 28, they also showed that VTA=>NAc neurons inhibit NAc PV interneurons. These interneurons have an inhibitory effect on MSNs. So this sentence needs to be qualified a little.

The sentences in lines 340-343 are hard to understand. I did not fully understand the point the authors were trying to make.

Lines 362-364: No mention of other 'VTA' drugs such as opioids and cannabinoids?

References:

References 4, 17, 20, 25 and 26 must be corrected or completed.

Reviewer #2

(Remarks to the Author)

In this work, Le Borgne and colleagues describe a well-designed series of experiments that investigate the role of two subpopulations of VTA dopamine neurons that project to the NAc and amygdala. These populations demonstrate a differential response to both nicotine and ethanol (activation vs. inhibition). The authors characterize the inhibition response to be the result of a GABAergic feedback loop. This is an important study as it increases our knowledge of the different subpopulations of VTA dopamine neurons that exhibit distinct roles depending on their projection target. There are several new discoveries here that will be very important for the nicotine field. Such as the finding that NAc-projecting DA neurons exhibit higher sensitivity than the Amg-projecting dopamine neurons. This clearly indicates that the nAChR subtypes in these distinct populations are different. This also supports the differential response (activation vs. inhibition) of these two projection pathways. Overall, this is strong body of work; but, there are some concerns that need to be addressed.

Major Concerns

- In Figure 2E, the authors state that the β 2NAcVEc frequency distribution exhibited a bimodal distribution to nicotine. The data clearly shows at least three populations of frequency distribution at -50%, 0%, and ~60%. The authors should adjust their designation or provide justification for why this was determined to be a bimodal result.
- Similarly, there may need to be further consideration for the data in Figure 2H.....
- It is not clear if the data provided in Figure 3E comes from 10Hz, 20Hz, or both stimulation frequencies. This should be clarified.
- There are two different statistical comparisons being made with the O-maze results in Figure 4. Despite this, the authors use the same type of asterisks and this makes the designation in Figures 4D and 4E difficult to determine the comparisons the authors are making. The authors should consider using different colors or different symbols to distinguish the comparisons between and within groups.
- In several experiments the authors employ their viral methods in distinct NAc areas (core, MSh, LSh). However, the authors largely refer to the NAc as a whole in their discussion. The authors should, to the best of their ability, discuss how their data informs us how these distinct VTA NAc and NAc VTA Amg pathways may be connected to specific NAc populations.
- The finding that the NAc-projecting DA neurons exhibit different sensitivities to nicotine when compared to the Amg-projecting neurons is intriguing. It would be beneficial to the readers if the authors could postulate what they would expect to be different in the nAChR subtypes/populations in these distinct groups. Do the authors think that nAChR-mediated desensitization may play a role in the population that is inhibited by nicotine application?
- The fact that both nicotine and ethanol produce a similar trend in these pathways is an important finding. Could the authors comment on how this finding may provide new insight into the high rates of co-use of nicotine and alcohol?
-

Minor Concerns

- In Figure 1 (C, D, H, etc.) the red and blue are clearly marked as nicotine or ethanol conditions that exhibit an increase or decrease in frequency. However, the saline traces are not marked in the figure. It would improve clarity if the grey traces were marked as saline groups as the readers may miss the notation in the figure legends.
- In Figure 1G and 1J, the two plots are not easily understood as to what each plot represents, especially if one were looking at a black/white printout. It may be useful to add labels to these plots.
- On page 7, line 191: There is a typo as the 61.6% of neurons for the NAcLSh is written to have a ratio of "8/1".
- Page 8, line 233: reverse should be "reversed"
- Page 7, Line 224, Supplementary Information: change "during a night" to "overnight".
- Page 8, line 242, Supplementary Information: The type of fluorescent microscope should be detailed in the methods section.
- Given the importance of the finding in Figure S5, the authors should consider fitting this data into the main paper (at least panels B-E).
- The WT and DATiCRE mice used in this study were all males. Could the authors add a justification or explanation for this to the methods section?

Version 1:

Reviewer comments:

Reviewer #1

(Remarks to the Author)

The authors responded very adequately to the comments, corrections and clarifications requested and added data that, overall, make this work even more impactful and beneficial in the field of nicotine and alcohol's impact on VTA function. One final minor improvement could be that To avoid confusion regarding the design of the experiments illustrated in Figs 4D-G, it may be useful to slightly update the panel 4D to clearly show the more medial positioning of the tetrodes and the more lateral positioning of the optical fiber.

Reviewer #2

(Remarks to the Author)

The authors have done an excellent job responding to the comments and critique I provided for the previous version of this manuscript. I have no additional substantial comments or critiques to add.

REVIEWER COMMENTS

Reviewer #1 (Remarks to the Author):

In their manuscript, Le Borgne et al. employ a broad range of cutting-edge techniques to compellingly reveal an inhibitory feedback circuit in ventral tegmental area (VTA) dopaminergic neurons projecting to the amygdala (DA VTA→Amygd), which induces anxiety following nicotine or alcohol intake. This circuit is activated via dopaminergic neurons projecting to the nucleus accumbens (DA VTA→NAc). Using in vivo juxtacellular single-neuron recordings combined with retrograde labeling, they demonstrate that both nicotine and alcohol induce activation or inhibition of DA neurons depending on their projection target—activation for neurons projecting to the NAc and inhibition for those projecting to the amygdala.

To better understand the inhibition of DA VTA→Amygd neurons, the authors employed patch-clamp techniques, showing that nicotine preferentially activates DA VTA→NAc neurons over DA VTA→Amygd neurons, independent of GABAergic interneurons in the VTA. Further, through selective rescuing of the beta2 nAChR's subunit in either VTA→NAc or VTA→Amygd neurons in beta2KO mice, as well as in DAT-Drd2 cKO mice, they excluded the involvement of intra-VTA GABAergic or intra-VTA dopaminergic circuits in nicotine's inhibitory effects. By combining optogenetics with patch-clamp recordings, Le Borgne et al. demonstrated that a GABAergic projection from the NAc to the VTA inhibits DA VTA→Amygd neurons. This pathway, activated by DA VTA→NAc neurons, was confirmed using optogenetic approaches alongside in vivo electrophysiological recordings. Finally, using loss-of-function methods with inhibitory opsins, the authors confirmed the necessity of this pathway for nicotine's inhibitory effect on VTA DA neurons and for the anxiogenic effects induced by both nicotine and alcohol. Together, these findings offer significant insights into the complex neural circuits involving VTA dopaminergic neurons, particularly in the context of addiction to nicotine and alcohol. Overall, this is a timely and high-quality study that is poised to make a substantial impact on the field of addiction neuroscience and the physiological understanding of VTA dopaminergic neurons. I have a few comments and suggestions regarding the work.

We would like to thank the reviewer for his/her thoughtful and encouraging feedback. We greatly appreciate the recognition of the study's contribution to the field of addiction neuroscience and to the understanding of VTA circuitry. We have revised the manuscript to address the different concerns raised, giving us the opportunity to significantly improve our work.

Abstract:

Line 26: The 'it' is kind of ambiguous. I guess it refers to nicotine.

We have replaced "it" with "nicotine".

Results:

A critical finding from the paper is that most of the recorded DA neurons respond with the same polarity to nicotine and ethanol. This finding is only illustrated in Fig S1D-E. I think it should appear in the main figure.

We completely agree, the results from Figure S1 C-D-E have been moved to Figure 1 C-F-G. We have also decided, for clarity, to split Figure 1 into two, with a new Figure 1 describing the responses of dopaminergic neurons to nicotine and ethanol injection, and a new Figure 2 focusing on NAc- and Amg- projecting neurons.

While most recorded cells respond to nicotine and ethanol with the same polarity, about 20% of them are nicotine-inhibited and ethanol-activated and none are found to be nicotine-activated and ethanol-inhibited. I think this finding should be highlighted and discussed especially as they represent roughly a similar fraction than nicotine-inhibited ethanol-inhibited neurons.

As advised by the reviewer, the population of neurons inhibited by nicotine but activated by EtOH has been highlighted in both the heat map (Figure 1F) and the location map (Figure 1H).

Regarding the anatomical location of neurons, we also added a new analysis (Figure 1H, color code) showing that EtOH-/Nic- neurons have a distinct mediolateral distribution compared to EtOH+/Nic+ neurons, suggesting that they may represent two distinct neuronal populations. In addition, EtOH+/Nic- neurons are not anatomically segregated from the other two populations. This suggests that, EtOH+/Nic- neurons may constitute a less lateralized third subpopulation. However, the drastic reduction of this population among identified NAc- and Amg- projecting neurons suggests that Nic-/EtOH+ neurons may project to another distinct brain region. This is now discussed in lines 97-103 and 130-132.

I think the author's claim related to the similarity in medio-lateral segregation between ethanol and nicotine responses would be reinforced if they directly compare the medio-lateral distribution of nicotine-activated neurons and ethanol-activated neurons. Same thing with nicotine-inhibited neurons and ethanol-inhibited neurons. As recommended by the reviewer, we provided the location map and distribution of neurons based on their responses to nicotine and ethanol independently (see Figure S1 C-G). We directly compared the mediolateral, anteroposterior, and dorsoventral distributions of nicotine-activated and -inhibited neurons with those of ethanol-activated and -inhibited neurons (see Figure S1 D-G).

In lines 107-111, it is not explicit it refers to nicotine injections.

References to nicotine intravenous injections are now explicitly stated (see lines 122).

Double labelling in Figure S2B and D is hard to see. Maybe add arrows to point double-fluorescence, split channels, or add zoom. Scale bars are missing in Figure S2B to D.

We added zoom and arrows to point the double-fluorescence, as well as scale bars (see Figure S2 B and D).

In line 130, the authors suggest a 'differential expression of nAChRs expression'. Is there any experimental evidence supporting this?

The differences in nicotinic currents observed between the two neuronal populations could result from different global levels of nAChR expression and/or a composition of nAChRs with distinct subunit stoichiometry in DA neurons projecting to the NAc and Amg. However, this question was not directly addressed in the present study, and there is no direct experimental evidence for differences in nAChR expression between these populations. These issues are now discussed in more detail in lines 345-356.

In lines 135-140, I'm not sure I understand the authors' rationale to explain the increase in the amplitude of sIPSC in NAc-projecting neurons.

First, because the changes in inhibitory postsynaptic currents were evoked by nicotine in these experiments, we decided to replace 'sIPSC' with 'IPSC' to avoid confusion. In the *ex vivo* experiments (Figure 3B–C), recordings were performed in coronal VTA slices, where most long-range inputs to the VTA are likely severed. In the absence of TTX, the observed changes in inhibitory postsynaptic currents (IPSCs) under nicotine application therefore primarily reflect the spiking activity of local GABAergic interneurons within the VTA. In NAc-projecting DA neurons, nicotine application led to a significant increase in the amplitude of IPSCs, without a corresponding change in their frequency. This increase in amplitude likely reflects a synchronization of GABAergic interneuron firing, driven by the activation of somatic nAChRs. Such synchronized input may result in larger postsynaptic events (amplitude), even if the overall number of events (frequency) remains unchanged—possibly due to temporal overlap and summation of individual IPSCs.

In contrast, Amg-projecting DA neurons did not exhibit any significant change in either IPSC amplitude or frequency following nicotine application. This suggests that nicotine selectively enhances GABAergic input onto NAc-projecting DA neurons, but not onto Amg-projecting neurons at least at the dose used (see lines 158-160).

To assess whether these differences could be attributed to basal differences in GABAergic synaptic organization, we recorded miniature IPSCs (mIPSCs) in the presence of TTX. No significant differences in amplitude or frequency were observed between the two populations under these conditions (Figure S3, see line 162-166). This indicates that baseline GABAergic connectivity is comparable, and that the nicotine-induced differences are likely due to active modulation of the local GABAergic network rather than structural differences in synaptic input (see line 167-169).

The design of Figure 3C feels confusing. Maybe an alternative could be to split it into pie charts for % connectivity and bar graphs for the current amplitude.

For clarity, we separately present the % connectivity with pie chart and for the current amplitude with bar graphs (see Figure 4C).

The difference in the proportion of VTA=>Amygd neurons responding to stimulation of NAc=>VTA neurons according to their location in the NAc (Fig 3C) is not discussed.

We agree, this is important to discuss this point. The differences in connectivity from NAc subregions onto BLA-projecting DA neurons are now discussed (see lines 388-397).

YFP labelling and co-expression with TH is hard to see in Figure 3D right panel. Also the scale bar is missing.

Double labeling has been highlighted, and scale bars have been added (see Figure 3D).

The authors should rename what they call 10 or 20 Hz optogenetic stimulation consistently throughout the text to make it clear that one is constant and the other is pulsed.

In line with the reviewer recommendation, we have renamed our 10 Hz and 20 Hz optogenetic stimulation protocols to “regular photostimulation” and “burst photostimulation,” respectively, throughout the manuscript. We have also updated the Methods section to include a more detailed explanation of each stimulation pattern.

Regarding optogenetic stimulations and recordings of DA neurons, I don't understand why excited ChR2-transfected DA neurons don't reach average frequencies close to the light stimulation patterns. Did the authors try some opto-tagging experiments to precisely separate ChR2-expressing neurons from the rest of DA neurons?

We would like to thank the reviewer for this comment. We attempted to identify light-responsive neurons that might show clear photo-tagging but we didn't find any. Specifically, none of the recorded neurons showed a significant increase in their probability of firing action potentials during the 10 ms following light pulse onset compared to the 10 ms between light pulses (see Figure S5B). This likely reflects suboptimal spatial alignment between the optic fiber and tetrodes (~300 μm lateral, ~100 μm ventral), which limited the efficiency of direct photostimulation—a limitation consistent with previous reports (see lines 245–249 and methods section lines 154-158). Nevertheless, the light-evoked inhibition in a separate group of neurons strongly suggests a network-level effect, likely resulting from ChR2 activation of NAc-projecting DA neurons. This has been clarified and discussed in the manuscript lines 249-252.

In Figure S5E (now Fig 5G), the effect observed after Opto+Nic looks higher in magnitude than the effect of Nic alone. Is this true? Also this figure shows a significant effect of saline injection. This should be discussed.

We would like to thank the reviewer for pointing this out. Although the nicotine-induced activation appears stronger under the Opto+Nic condition compared to Nic alone in Figure S5E, this difference is not statistically significant (paired t-test, $p=0,08$).

Regarding the significant effect of saline injections, three of the four recorded neurons received a saline injection and showed a decrease in firing rate post-injection, which is not uncommon. Saline injections can sometimes elicit their own responses or coincide with spontaneous changes in firing patterns. As explained in the Methods, our

primary analysis of drug responses focuses on changes relative to baseline activity, and our secondary analysis also compares the maximum firing changes observed after saline with those after drug injections. Still, the saline injection effect might interfere with any nicotine-induced changes, but it is taken into account in our analysis. Overall, given the small sample size and the potential confounding influence of saline, drawing definitive conclusions about the possible modulation of nicotine-induced activation by the NAc-VTA projection based on these experiments is challenging and requires further investigation, as discussed in lines 277–278. However, the NAc-VTA projection does not appear to be necessary for nicotine to trigger DA neuron activation.

In Figure S6 the authors should also display the effects on general locomotion of ethanol and optogenetic inhibition in the EOM (like figure S6C for nicotine)

As requested by the reviewer, these data are now presented in Figure S6F and replace the time spent in the open arms, which has been moved to the main Figure 5F.

(Minor)

For the consistency between panels, please change the order of pie charts in Fig 1L.

This has been done.

Figure S2D could be modified to also separate Nic-/EtOH+ neurons.

We believe that the reviewer was referring to the heat map shown in the previous Figure S1D. Nic-/EtOH+ neurons are now separated in the heat map shown in Figure 1F (see the reviewer's first comment).

The distribution histograms in Fig S1F and S2E should be revised as the overlay between bars is unclear.

All histogram distributions have been revised in the manuscript to avoid overlay.

Horizontal axis labels are missing in left panels in Fig S1F and S2F

This has been corrected.

In Figures 2A-C, and S3, please keep the same order between NAc-DA and BLA-DA neurons.

This has been modified.

Line 191: n = 8/13 and not n = 8/1.

We apologize for this error, which has been corrected.

Discussion:

Line 276: I'm not sure I would call anxiety and reward 'opposite behaviors'

We agree with the reviewer's comment and have changed it to "distinct affective behaviors".

Lines 305 to 311- of the discussion is highly (too?) speculative.

Indeed:

Lines 305-307: There is no evidence that the abundance of the beta2 subunit varies according to the particular type of DA neuron. The hypothesis of different combinations of beta2*nAChR receptors is an undiscussed alternative. This hypothesis is now presented (see line 346) and thoroughly discussed (see lines 347–362).

Lines 307-309: This sentence is not clear. As rescuing is specific to VTA=>NAc neurons, other VTA neurons and in particular VTA=>Amygdala neurons do not have beta2 de facto. Or are you implying that in a control mouse the VTA-Amyd neurons would have no beta2?

We reference neurons in B2 KO mice that did not re-express the beta2 subunit, including VTA-amygdala neurons. The sentence has been clarified (see lines 330-332).

Lines 309-311: In reference 28, they also showed that VTA=>NAc neurons inhibit NAc PV interneurons. These interneurons have an inhibitory effect on MSNs. So this sentence needs to be qualified a little.

Indeed, the work of Sun and Laviolette shows that intra-VTA infusion of nicotine exerts opposing effects on the MSN and PV neurons of the NAc. Rewarding doses of nicotine activate MSN neurons while inhibiting NAc PV neurons, whereas aversive doses have the opposite effect on these NAc neuronal populations. The sentence has now been qualified to refer specifically to the rewarding dose of nicotine (see lines 332-334).

The sentences in lines 340-343 are hard to understand. I did not fully understand the point the authors were trying to make.

We believe that previous studies using targeted optogenetic modulation of specific NAc subregions (Qi et al. 2022, Yang et al. 2018) may not fully capture the global and synchronous activation of the entire VTA-NAc projection system that occurs during nicotine exposure. Our study simulates this broader activation pattern, which could explain the observed differences in anxiety modulation. (see now lines 381-387).

Lines 362-364: No mention of other 'VTA' drugs such as opioids and cannabinoids?

We have added a section discussing drugs that act at the somatic level of the VTA (lines 409-412).

References:

References 4, 17, 20, 25 and 26 must be corrected or completed.

We apologize for these errors, which have been corrected.

Reviewer #2 (Remarks to the Author):

In this work, Le Borgne and colleagues describe a well-designed series of experiments that investigate the role of two subpopulations of VTA dopamine neurons that project to the NAc and amygdala. These populations demonstrate a differential response to both nicotine and ethanol (activation vs. inhibition). The authors characterize the inhibition response to be the result of a GABAergic feedback loop. This is an important study as it increases our knowledge of the different subpopulations of VTA dopamine neurons that exhibit distinct roles depending on their projection target.

There are several new discoveries here that will be very important for the nicotine field. Such as the finding that NAc-projecting DA neurons exhibit higher sensitivity than the Amg-projecting dopamine neurons. This clearly indicates that the nAChR subtypes in these distinct populations are different. This also supports the differential response (activation vs. inhibition) of these two projection pathways. Overall, this is strong body of work; but, there are some concerns that need to be addressed.

We thank the reviewer for their positive assessment of our work and for recognizing its importance in understanding the distinct roles of VTA dopamine neuron subpopulations in relation to nicotine and alcohol addiction. We have revised the manuscript to address the different concerns raised, giving us the opportunity to significantly improve our work.

Major Concerns

- In Figure 2E, the authors state that the β 2NAcVEc frequency distribution exhibited a bimodal distribution to nicotine. The data clearly shows at least three populations of frequency distribution at -50%, 0%, and ~60%. The authors should adjust their designation or provide justification for why this was determined to be a bimodal result.

We thank the reviewer for raising this point. We agree that the data in Figure 2E (now 3E) do not strictly fit a bimodal distribution but rather suggest at least three distinct peaks, corresponding to excitatory, inhibitory, and no responses. This multimodal pattern likely reflects varying levels of β 2 expression among NAc-projecting neurons. We have adjusted our terminology accordingly and expanded our discussion of these findings (see lines 185 and 186-188). A similar designation adjustment has been made for Figure 3H (see below) and Figure 4E (see line 242).

- Similarly, there may need to be further consideration for the data in Figure 2H.

Accordingly, we also adjust the designation of the distribution in Figure 3H (see lines 191-194).

- It is not clear if the data provided in Figure 3E comes from 10Hz, 20Hz, or both stimulation frequencies. This should be clarified.

We apologize for the confusion; the data come from 10Hz stimulation, which is now clearly indicated in the text (see line 243) and Figure 3E.

- There are two different statistical comparisons being made with the O-maze results in Figure 4. Despite this, the

authors use the same type of asterisks and this makes the designation in Figures 4D and 4E difficult to determine the comparisons the authors are making. The authors should consider using different colors or different symbols to distinguish the comparisons between and within groups.

As suggested by the reviewer, we used different symbols to distinguish the comparisons within (*) and between (#) groups concerning the O-maze results. This is now indicated in figure legends.

- In several experiments the authors employ their viral methods in distinct NAc areas (core, MSh, LSh). However, the authors largely refer to the NAc as a whole in their discussion. The authors should, to the best of their ability, discuss how their data informs us how these distinct VTA \rightarrow NAc and NAc \rightarrow VTA \rightarrow Amg pathways may be connected to specific NAc populations.

In line with the reviewer recommendation, we highlight that VTADA-BLA neurons, located in the medial VTA (Nguyen et al. 2021 and present study) and involved in encoding motivational salience (Lutas et al. 2019, Sias et al. 2024), preferentially receive GABAergic inputs from the NAc Core, an NAc subregion involved in motivational salience signaling (Kutlu et al. 2021). These connections further support the idea of functional lateralization within the VTA, where specific dopaminergic circuits encode motivational salience versus valence (see line 388-397).

- The finding that the NAc-projecting DA neurons exhibit different sensitivities to nicotine when compared to the Amg-projecting neurons is intriguing. It would be beneficial to the readers if the authors could postulate what they would expect to be different in the nAChR subtypes/populations in these distinct groups. Do the authors think that nAChR-mediated desensitization may play a role in the population that is inhibited by nicotine application?

We thank the reviewer for this interesting point of discussion that was missing in the manuscript. Differences in nAChR subtypes according to the subpopulation, as well as the implications of nAChR-mediated desensitization, have been discussed in lines 347-362.

- The fact that both nicotine and ethanol produce a similar trend in these pathways is an important finding. Could the authors comment on how this finding may provide new insight into the high rates of co-use of nicotine and alcohol?

We agree with the reviewer that our results should be discussed in the context of co-administration of the two drugs, as they suggest that the two drugs may have a synergistic effect on DA pathways. However, experiments in which the two drugs were injected simultaneously were critically lacking in the study to directly address this point. Therefore, we have decided to provide new data comparing the effects of nicotine and ethanol alone with the co-administration of both drugs on the activity of VTA DA neurons (see methods section 119-121). These data, presented in Figure S1J, show that nicotine and ethanol have a cumulative effect on cell activity, demonstrating that the processes by which the drugs act are neither counteracting nor saturating, but rather cooperative and complementary (see lines 108-111). In the light of these results, we now discuss in detail how our findings might provide insight into the high prevalence of co-use of nicotine and alcohol (see lines 416-433).

Minor Concerns

- In Figure 1 (C, D, H, etc.) the red and blue are clearly marked as nicotine or ethanol conditions that exhibit an increase or decrease in frequency. However, the saline traces are not marked in the figure. It would improve clarity if the grey traces were marked as saline groups as the readers may miss the notation in the figure legends.

As suggested, grey traces have been marked as saline.

- In Figure 1G and 1J, the two plots are not easily understood as to what each plot represents, especially if one were looking at a black/white printout. It may be useful to add labels to these plots.

As suggested, labels have been added to these panels, which are now part of the new Figure 2, based on the recommendation of another reviewer.

- On page 7, line 191: There is a typo as the 61.6% of neurons for the NAcLSH is written to have a ratio of "8/1".

We apologize for this error, which has been corrected (8/13).

- Page 8, line 233: reverse should be "reversed"

It has been corrected line 270.

- Page 7, Line 224, Supplementary Information: change "during a night" to "overnight".

It has been corrected line 238.

- Page 8, line 242, Supplementary Information: The type of fluorescent microscope should be detailed in the methods section.

It has been added in the methods section line 256.

- Given the importance of the finding in Figure S5, the authors should consider fitting this data into the main paper (at least panels B-E).

We agree that the data presented in Figure S5 are important; however, for the sake of clarity in Figures 4 and 5, we prefer to keep them as supplementary material. The data presented in Figure S5A (now A-C) relate to nicotine injections, while Figure 4 focuses on optogenetic stimulation of a specific DA neuron subpopulation. Similarly, the data in Figure S5D-E (now D-G) show the effect of NAc projection inhibition on nicotine-induced activation of DA neurons, whereas Figure 5 focuses on the role of NAc projection in nicotine-induced inhibition and its behavioral consequences.

- The WT and DATiCRE mice used in this study were all males. Could the authors add a justification or explanation for this to the methods section?

Following justification have been added in the methods section (see methods section lines 12-21):

“The decision to use only one sex in WT mice was based on the nature of the experiment, which required juxtacellular labeling of recorded neurons identified by retrograde tracing. This method yields a low number of double-labeled neurons per animal when using retrobeads and neurobiotin, so a large number of animals is required to obtain sufficient data. Splitting the data by sex would make post hoc statistical comparisons between sexes impractical unless an even larger cohort was used. The use of a single sex in DAT-Cre mice for in vivo recordings combined with optogenetics, which involves a limited number of recordings per animal, was justified for the same reason. Finally, we chose to establish the correlation between electrophysiology and behavior by testing mice of the same sex in both paradigms.”

REVIEWER COMMENTS

Reviewer #1 (Remarks to the Author):

The authors responded very adequately to the comments, corrections and clarifications requested and added data that, overall, make this work even more impactful and beneficial in the field of nicotine and alcohol's impact on VTA function.

One final minor improvement could be that To avoid confusion regarding the design of the experiments illustrated in Figs 4D-G, it may be useful to slightly update the panel 4D to clearly show the more medial positioning of the tetrodes and the more lateral positioning of the optical fiber.

We thank the reviewer for their positive assessment of our work. As suggested, we have more clearly indicated the relative position of the tetrode and the optical fiber in panel 4D.

Reviewer #2 (Remarks to the Author):

The authors have done an excellent job responding to the comments and critique I provided for the previous version of this manuscript. I have no additional substantial comments or critiques to add.

We thank the reviewer for their positive assessment of our work.